# Nitrogen and Nod factor signaling determine *Lotus japonicus* root exudate composition and bacterial assembly

Ke Tao[1,5], Ib T. Jensen [1,6], Sha Zhang[1], Eber Villa-Rodríguez[1], Zuzana Blahovska[1], Camilla Lind Salomonsen [2], Anna Martyn [1,7], Þuríður Nótt Björgvinsdóttir[2], Simon Kelly[1,8], Luc Janss[3], Marianne Glasius [2], Rasmus Waagepetersen[4] & Simona Radutoiu [1] ✉

Symbiosis with soil-dwelling bacteria that fix atmospheric nitrogen allows legume plants to grow in nitrogen-depleted soil. Symbiosis impacts the assembly of root microbiota, but it is unknown how the interaction between the legume host and rhizobia impacts the remaining microbiota and whether it depends on nitrogen nutrition. Here, we use plant and bacterial mutants to address the role of Nod factor signaling on *Lotus japonicus* root microbiota assembly. We find that Nod factors are produced by symbionts to activate Nod factor signaling in the host and that this modulates the root exudate profile and the assembly of a symbiotic root microbiota. *Lotus* plants with different symbiotic abilities, grown in unfertilized or nitrate-supplemented soils, display three nitrogen-dependent nutritional states: starved, symbiotic, or inorganic. We find that root and rhizosphere microbiomes associated with these states differ in composition and connectivity, demonstrating that symbiosis and inorganic nitrogen impact the legume root microbiota differently. Finally, we demonstrate that selected bacterial genera characterizing state-dependent microbiomes have a high level of accurate prediction.

Plant tissues are habitats for communities of microbes, generally called the plant microbiota[1–3]. Members of these communities may colonize the intra- or intercellular space of the plants, as well as the surface of aerial and belowground organs[3–6]. Microbial colonization of plant tissues involves an extensive selection process. Evidence for an active role of the host in this process has emerged from studies across numerous plant species and growth conditions[7–11], but the genetic mechanisms employed by the host to control microbiota assembly are largely unknown. Phylogenetically distinct plant species host different microbial communities, which primarily differentiate at a low taxonomic level[10,12,13]. Reconstitution experiments using synthetic communities revealed that bacterial isolates assigned to the same family and originating from the same soil may have preferences for colonizing the roots of legumes or brassica[12]. This host preference was observed only in the presence of metabolically active roots, suggesting that microbes adapt to plant-derived environmental niches. How different plants use their specific metabolites to modulate microbiota or only select members of it is important for our basic understanding of microbiota assembly, as well as for applied strategies aiming to harness beneficial plant–microbe associations for a sustainable increase of plant yield and resilience. Plant species differ in root morphology and cell wall composition, as well as the pattern and magnitude of

[1]Department of Molecular Biology and Genetics, Aarhus University, Aarhus, Denmark. [2]Department of Chemistry, Aarhus University, Aarhus, Denmark. [3]Center for Quantitative Genetics and Genomics, Aarhus University, Aarhus, Denmark. [4]Department of Mathematical Sciences, Aalborg University, Aarhus, Denmark. [5]Present address: Department of Biology, University of Copenhagen, Copenhagen, Denmark. [6]Present address: Department of Mathematical Sciences, Aalborg University, Aarhus, Denmark. [7]Present address: Department of Plant-Microbe Interactions, Max-Planck-Institute for Plant Breeding Research, Cologne, Germany. [8]Present address: Biotechnology, Lincoln Agritech, Canterbury, New Zealand. ✉e-mail: radutoiu@mbg.au.dk

metabolites secreted in the rhizosphere, which in turn impact the associated microbiota[14–19]. These traits also vary within the same species depending on plant genotype or physiological state. The nutritional state of the host was consistently identified to impact microbiota composition across plant species[20,21]. In nutrient-depleted states, plants adjust the profile of secreted metabolites which leads to the enrichment of specific members from soil microbiota[22–24], some of which engage in symbiotic associations with the host[25–27].

Legume plants are genetically equipped to enrich their rhizosphere with soil bacteria that fix atmospheric nitrogen which are accommodated inside their roots and root nodules[25,27,28] and can thus overcome the lack of nitrogen in the soil. Nitrogen-fixing symbiosis is a facultative trait. Legumes initiate symbiotic signaling and allow bacteria to infect their roots if nitrogen in the soil is insufficient for plant growth. Most of the symbionts do not fix nitrogen as free-living bacteria in the soil, and legume plants select them based on complex signal exchange and recognition[29–31]. Consequently, even bacterial mutants and isolates with impaired or reduced capacity to fix nitrogen can colonize legume roots and nodules efficiently if they are able to produce compatible symbiotic signals, i.e. Nod factors[32,33], although the host plant receives little benefit from this exchange. Host-specific flavonoids are released from roots into the soil and, if recognized by symbionts, induce expression of bacterial genes for Nod factor biosynthesis[34,35]. Nod factors with bacteria-specific decorations are recognized by plant receptors localized on the plasma membrane[30,31,33,36]. The NFR1 and NFR5 receptors initiate symbiosis signaling in *Lotus japonicus* roots, thus triggering nodule organogenesis and infection thread formation[29,37]. Symbionts infect the roots and nodules via infection threads[5,38], where they continue to produce Nod factors while the infection threads progress inwards across plant root-cell layers and within nodule primordia[28,39]. Studies based on plants and bacterial mutants revealed that Nod factor-dependent signaling is necessary not only for the initiation of symbiosis but also for efficient bacterial infection and successful symbiotic association. In *Lotus*, the epidermal Nod factor receptor NFRe aids the symbiosis initiated by NFR1 and NFR5, enabling optimal signaling and symbiosis establishment[40], while the Chitinase *Chit5* cleaves bacterial Nod factors, maintaining an optimal level of signaling during infection which is necessary for infection thread progression inside nodule primordia[41].

Previous studies revealed that the host pathway required for symbiosis with nitrogen-fixing bacteria has a major impact on the root-microbiota assembly of legume plants[7,42–45], an effect that was retained in the presence of nitrogen-replete soil[7]. Root nodule symbiosis is a complex biological problem involving thousands of plant and bacterial genes, complex signaling, and metabolic reprogramming. However, it is unknown which components of symbiosis affect the structure of legume root microbiota, or if nitrogen nutrition impacts the structure of legume root microbiota remains unknown. Root-nodule symbiosis is an energy-demanding process for the host, and it is inhibited in nitrogen-replete conditions[46]. Therefore, when considering nitrogen nutrition, legumes can be found in three distinct states: starved, replete due to nitrogen-fixing symbiosis, or replete due to the uptake of inorganic nitrogen from the soil. Both forms of nitrogen-replete states ensure efficient plant growth and seed production, but legumes differ in the amount of nitrogen they require for high seed yield[47,48]. Legumes and their symbionts thus provide an appropriate system for investigating whether the acquisition of nitrogen through either symbiosis with a soil microbe or directly from the soil reserves has an impact on microbiota assembly, as well as how symbiosis signaling and nitrogen source contribute to the establishment of root microbiota.

Here, we use *Lotus japonicus* and *Mesorhizobium loti* mutants to address these questions. Experiments using unfertilized or nitrate-supplemented soil that is permissive or suppressive, respectively, for symbiosis identified that nitrogen nutrition and Nod factor signaling are major drivers of *Lotus* root microbiota. These findings were further

confirmed by results from studies based on gnotobiotic settings with synthetic communities. We found that root microbiota is dependent on nitrogen nutrition and source and that the bacterial genera characterizing these states have high prediction power. Importantly, our results based on Nod factor signaling in *Lotus* provide evidence that interactions established between the host and distinct members of soil microbiota can lead to a feedback effect on the remaining members of the community that is orchestrated by the host.

## Results

### The physiological state of *Lotus* impacts the associated bacterial communities

To investigate if the acquisition of nitrogen through symbiosis or from the soil has a different impact on the assembly of bacterial communities of *Lotus*, we grew wild-type plants (Gifu) and three mutants, *nfr5*, *nfre*, and *chit5*, in native, unfertilized soil, where either symbiosis is enabled or where the soil is supplemented with nitrate during the experiment (Fig. 1a). Analyses based on gnotobiotic, nitrogen-free conditions showed that these three mutants are impaired in Nod factor signaling at different stages of symbiosis[29,40,41]. Mutants in *Nfr5* do not form nodules or infection threads[29,37], those in *Nfre* form fewer but still functional nitrogen-fixing nodules[40], while *chit5* forms nodules that are non- or poorly infected[41]. Here, we observed that the shoot and nodulation phenotypes reported for these genotypes grown in gnotobiotic conditions were reproduced when plants were grown in unfertilized soil (Fig. 1b, d, e) and that nitrate fertilization inhibited symbiosis but ensured plant nutrition independent of symbiosis impairment (Fig. 1c, f).

Next, we analyzed the soil and plant-associated bacterial communities using 16S rRNA amplicon sequences. As expected, the diversity analysis of unplanted soil, rhizosphere, root, and nodules showed separation between the four compartments (Supplementary Fig. 1b), suggesting different selection pressures of the related bacterial community composition. Nitrate fertilization reduced the α-diversity of these communities in the unplanted soil, as well as in the rhizosphere of *nfr5*, while the remaining samples maintained a similar within-sample diversity (Fig. 2a and Supplementary Fig. 1a). Analyses of β-diversity using principal coordinate analysis (PCoA) with Bray-Curtis distances showed a clear separation of communities from our data set into three clusters. One contained communities from Gifu and *nfre* grown in unfertilized soil supported by nitrogen-fixing symbiosis. The second contained communities of nitrogen-starved plants: *chit5* and *nfr5*. The third contained communities of all genotypes grown in fertilized soil (Fig. 2b, c, Supplementary Fig. 1). These differences in bacterial communities followed the observed phenotypes (Fig. 1b, c) and the physiological state of the host, nitrogen-starved, symbiotically active, and nitrate-fertilized plants.

These results from analyses of α- and β-diversity show that both the nutritional state (nitrogen depleted or replete) and the source of nitrogen (symbiotic or inorganic) are major factors differentiating bacterial communities in soil-grown *Lotus* plants. Importantly, bacterial communities associated with symbiotically active plants differ from those associated with plants fertilized with inorganic nitrogen, even if plants are in a nitrogen-replete state in both conditions.

### Mutations in Nod factor signaling have a differential impact on the root microbiome

To determine if differences observed at the community level are associated with changes in the relative abundance (RA) of specific taxa, we clustered the 16S rRNA amplicons into ASVs that were assigned to a taxonomic rank[49]. Comparative analyses of the relative abundances (RA) at the order level showed no significant differences between Gifu and these symbiotic mutants in each of the two growth conditions (Fig. 2d, e and Supplementary Fig. 2c, d). Then, we investigated if significant differences are detectable at a lower taxonomic level. We

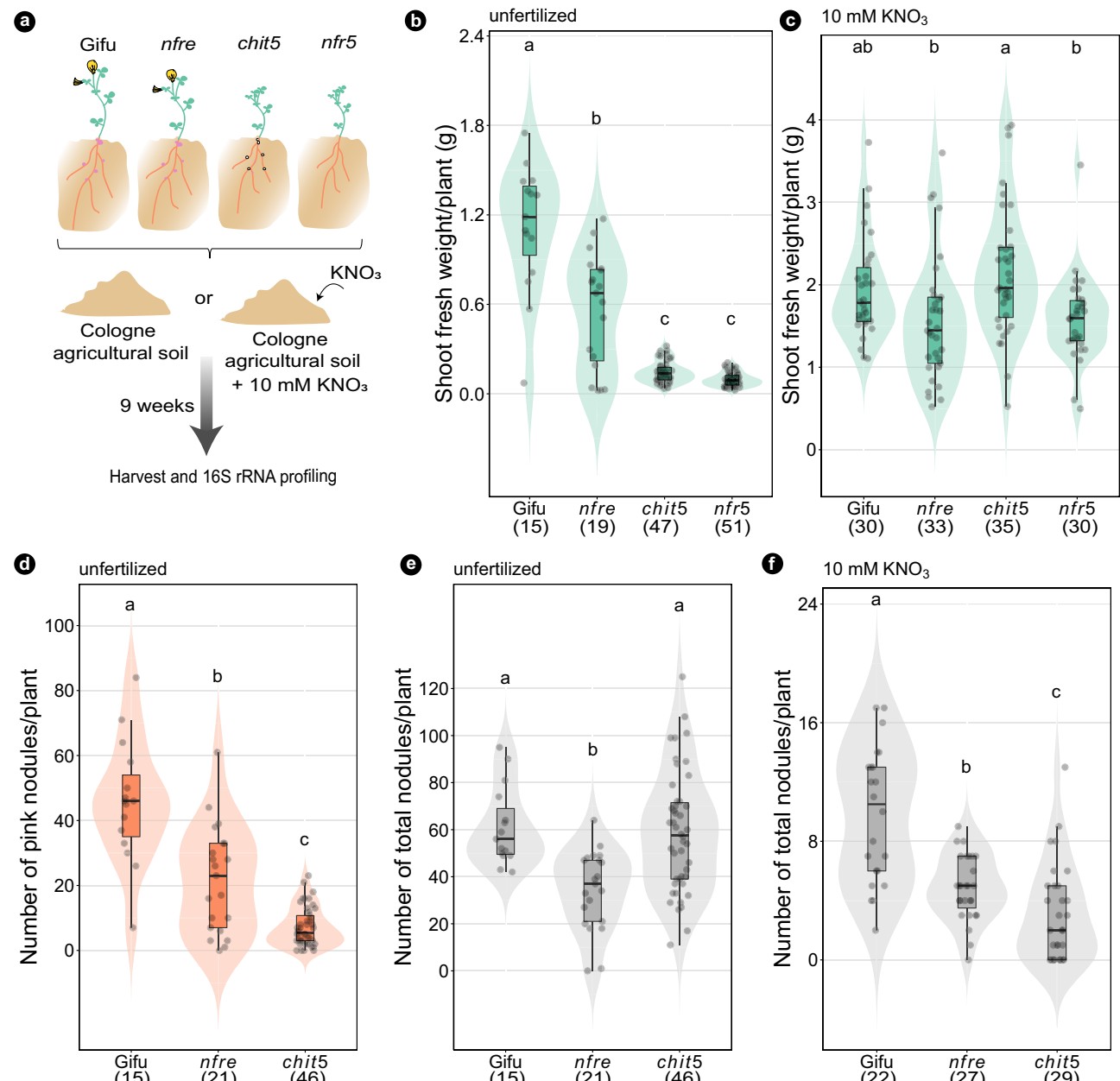

**Fig. 1 | Wild-type and Nod factor signaling mutants grown in soil show symbiosis-defective phenotypes compensated for by nitrate addition to the soil. a** Experimental design. Shoot fresh weight per plant of wild type, *nfre*, *chit5*, and *nfr5* grown in unfertilized Cologne soil (**b**) or soil supplemented with 10 mM KNO₃ (**c**). The number of pink nodules (**d**) and the total number of nodules (**e**) per plant of wild type, *nfre*, and *chit5* grown in unfertilized Cologne soil. The total number of nodules (**f**) per plant of wild type, *nfre*, and *chit5* grown in Cologne soil supplemented with 10 mM KNO₃. Each dot represents a value for individual plants. The number of analyzed plants are shown in brackets. The shape of the violins illustrates the density of the data in the analysed samples. Boxplots within the violin plots show the median, 75th percentile, and 25th percentile data sets. Letters within the plots (**b–f**) indicate statistically significant differences (Tukey HSD test, $p < 0.05$).

focused primarily on 141 ASVs with RA greater than 0.3% in Gifu roots (Fig. 3). Together, these have an average cumulative RA greater than 60% (Fig. 3b) and are assigned to 10 orders and 14 families, with the majority being members of Rhizobiales, Caulobacterales, and Burkholderiales (Fig. 3b and Supplementary Data 1). Communities of *nfre* were found to be most similar to Gifu; only 11 ASVs were significantly reduced (Fig. 3d and Supplementary Data 1). By contrast, 31 ASVs had a significantly reduced abundance in *chit5* and 48 ASVs in the *nfr5* roots. Most importantly, these ASVs belong to families that are highly abundant in Gifu roots such as *Oxalobacteraceae*, *Comamonadaceae*, *Rhizobiaceae*, and *Xanthobacteraceae* (Supplementary Data 1). ASVs belonging to *Mesorhizobium*, *Oxalobacteraceae*, *AsCoM_f_4827*

(Gammaproteobacteria), together accounting for more than 15% abundance in Gifu, were significantly reduced in *chit5* and *nfr5* (Fig. 3c, d, Supplementary Fig. 3, Supplementary Data 2). These differences at the ASV level can explain the clear separation of communities associated with starved (*nfr5* and *chit5*) versus symbiotic (Gifu and *nfre*) plants identified by the β-diversity and PERMANOVA analyses, where a significant effect of plant genotype on root communities was revealed (Supplementary Fig. 1f). Moreover, we also identified several ASVs whose abundance differ between wild type and mutants when plants were fertilized with nitrate (Supplementary Figs. 4 and 5), but these were not statistically significant. Together, these results show that a gradual impairment of Nod factor signaling has an impact

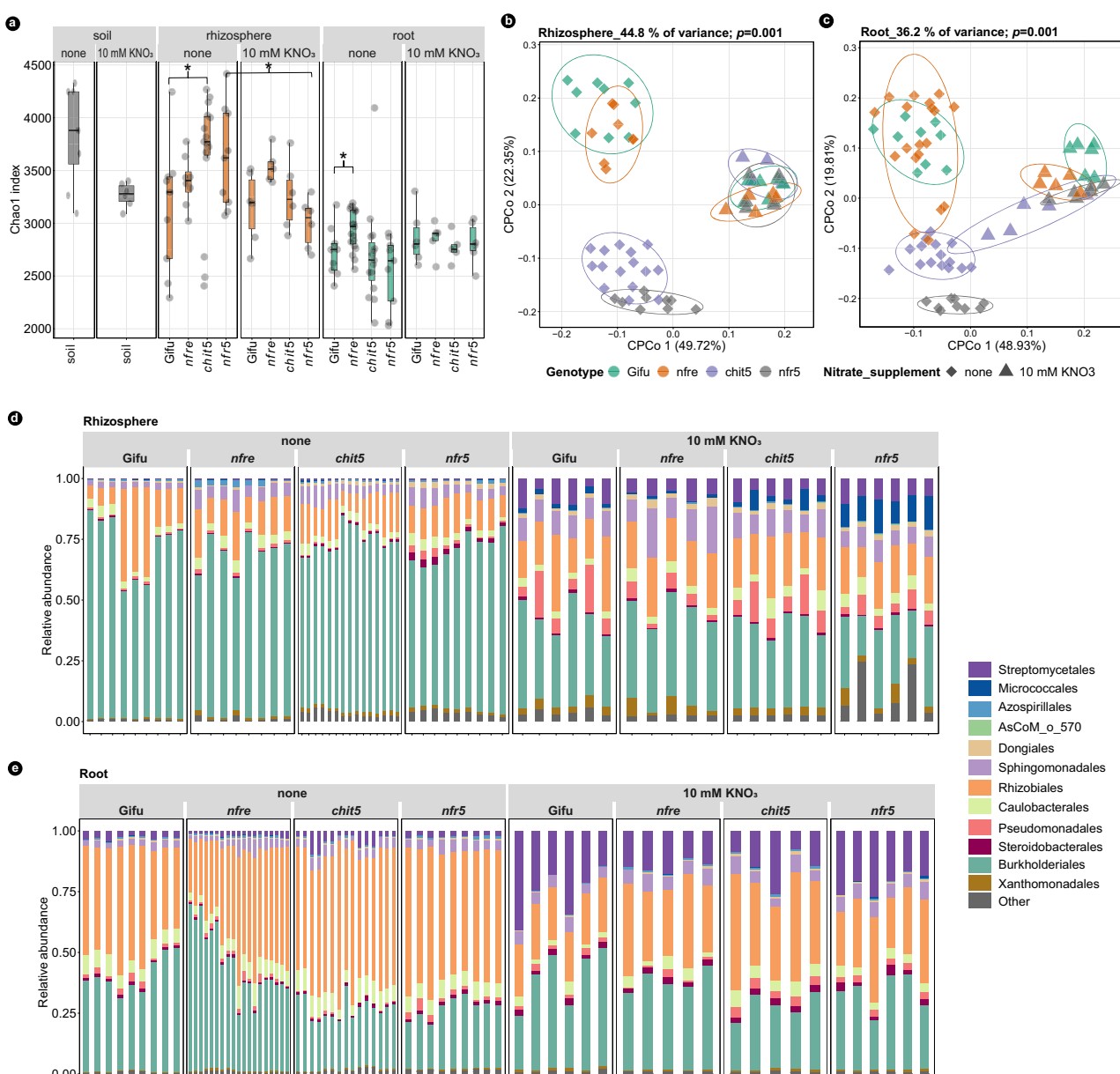

**Fig. 2 | Nitrate supplementation changes the soil, root, and rhizosphere community structures. a** Alpha diversity by Chao1 index for soil, rhizosphere, and root compartments of Gifu, *nfre*, *chit5*, and *nfr5*. **b** Constrained PCoAs of rhizosphere and root (**c**) communities show that Nod factor signaling and nitrate supplementation have a major effect on the associated bacteria. The analysis is constrained by both genotype and nitrate application. **d** Relative abundance of bacterial families for rhizosphere and root (**e**) samples. Boxplots of alpha diversity show the median, 75th percentile, and 25th percentile data sets. Asterisks within the alpha diversity plot indicate statistically significant differences between the two genotypes (Mann–Whitney U test, $p < 0.05$, $n = 3$ biological samples each analysed in 3 technical replica for unfertilized condition, $n = 3$ biological samples analysed in 2 technical replica for 10 mM KNO₃ fertilized condition). Columns indicate the replica and colors indicate the taxonomic assignment.

on root microbiota and that host genotype affects the composition of bacterial communities present in *Lotus* roots grown in symbiosis-permissive soil.

## Nitrate fertilization of the soil affects root and rhizosphere microbiomes

Next, we performed comparative analyses of bacterial communities under the two growth conditions and identified that the application of nitrate had a significant effect on bacterial communities: ASVs from 13 families had a reduced abundance in fertilized soil compared to unfertilized, unplanted soil (Supplementary Figs. 6–9, Supplementary Data 3). This explains the observed reduction in the α-diversity of the unplanted soil samples (Fig. 2a and Supplementary Fig. 1a).

Surprisingly, we found that nitrate fertilization had a much larger impact on the rhizosphere and root communities of all plant genotypes and that those ASVs affected by nitrate in the unplanted soil represented only a minor fraction of the overall changes detected in the root and rhizosphere compartments (Supplementary Figs. 6–9). The three major bacterial orders, Burkholderiales, Rhizobiales, and Streptomycetales, were all significantly affected in their RAs (Fig. 2d, e, Supplementary Fig 2). Burkholderiales, which dominated the rhizosphere of plants grown in unfertilized soil, was substantially reduced under fertilized soil conditions (Fig. 2d and Supplementary Fig. 2c), where the abundance of Pseudomonadales, Streptomycetales, and Sphingomondales increased (Fig. 2d and Supplementary Fig. 2c). Rhizobiales, which dominated the root communities of plants grown in

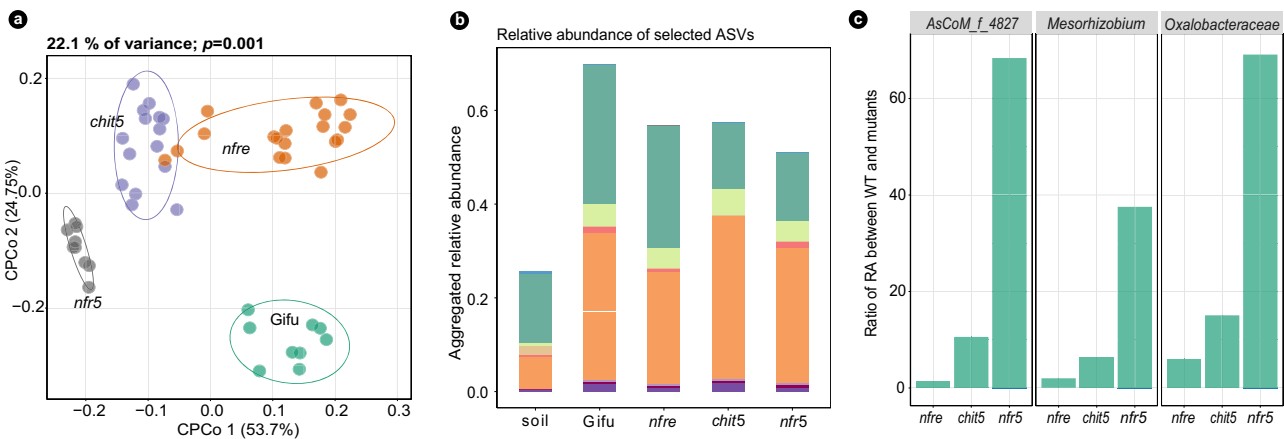

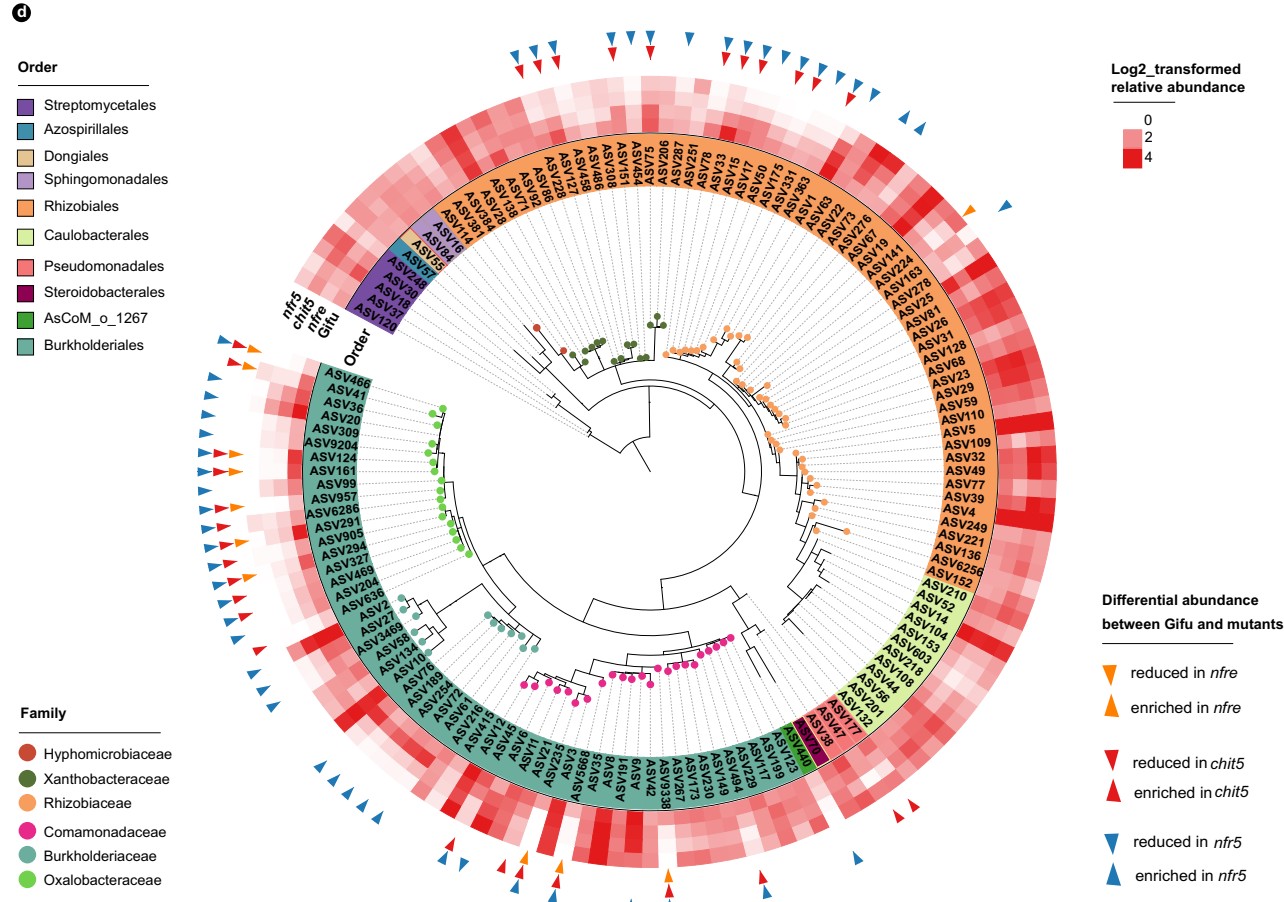

**Fig. 3 | Nod factor signaling contributes to root-associated microbiota of *Lotus* plants grown in unfertilized Cologne soil. a** Constrained PCoA of communities associated with roots of wild-type, *nfr5*, *nfre*, and *chit5*. **b** Cumulative relative abundance of selected ASVs (RA > 0.3% in roots of Gifu) in soil and roots of the four genotypes. **c** The ratio of RA between Gifu and mutants of the top three taxa in the roots based on selected ASVs: RA > 0.3% in roots of Gifu. **d** Distinct ASVs have a significantly different RA in mutant roots compared to wild-type Gifu. Selected

ASVs are presented in a phylogenetic tree constructed on the basis of 16S rRNA V5–V7 region. The taxonomic information is shown by color on the name of the ASV (order) and by color on the tree branch (family). The heatmap shows the log2-transformed RA of each ASVs in the roots of Gifu, *nfre*, *chit5*, and *nfr5* plants. Triangles on the outer layer of the heatmap point out ASVs that have a significantly different abundance compared to wild-type plants.

unfertilized soil (Fig. 2e and Supplementary Fig. 2d), was substantially reduced and was compensated for by an increase in the abundance of Streptomycetales. In addition to these changes in the cumulative abundances of major bacterial orders, the nitrate had a clear effect on the abundance of individual ASVs. As a consequence, the composition

of ASVs within the same families changed, indicating a possible shift in the functional capacities at the family level (Supplementary Figs. 6–9).

Together, these results show that nitrate fertilization has a consequential impact on the rhizosphere and root microbiomes of *Lotus*,

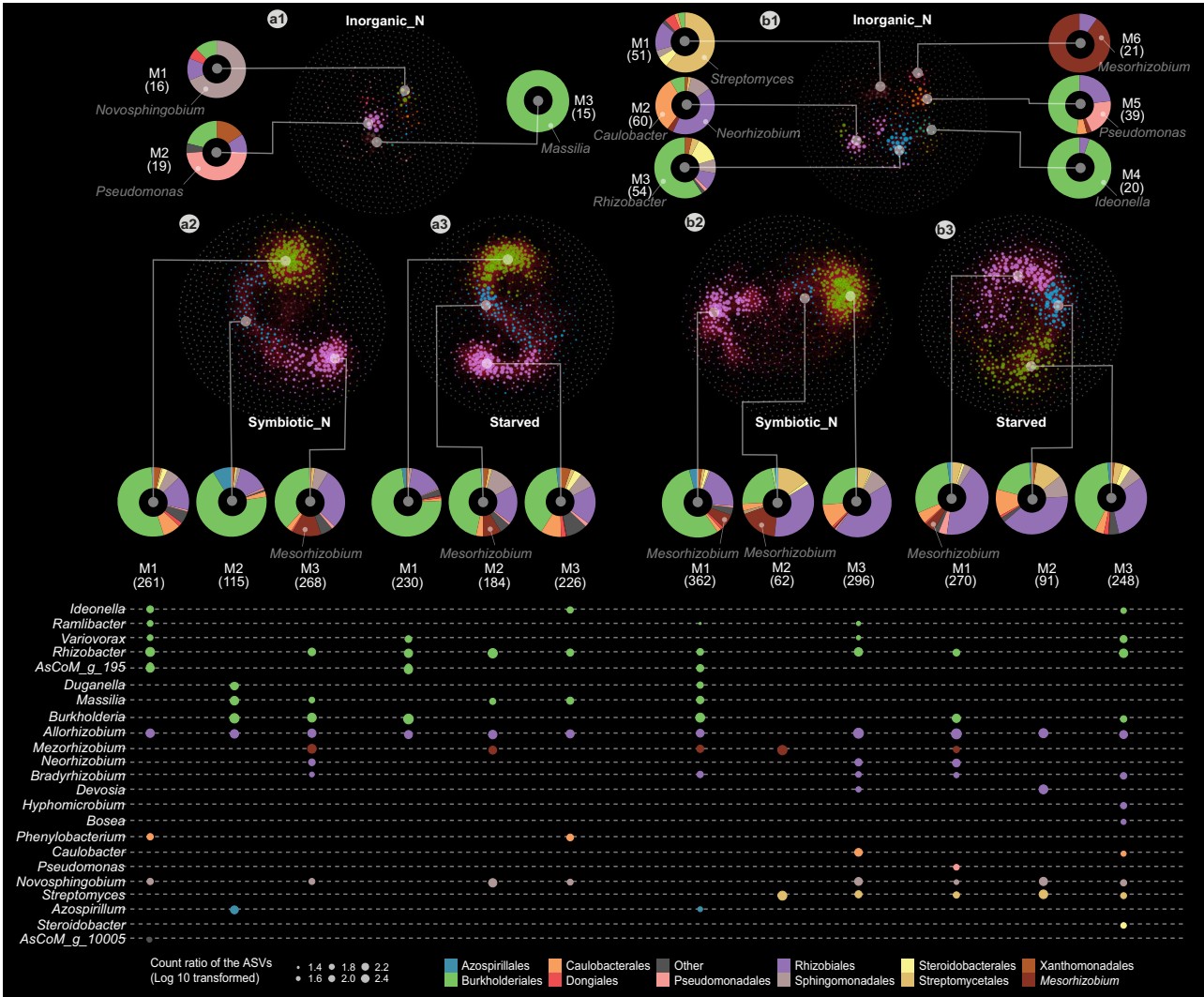

**Fig. 4 | Nitrogen nutritional status and source drive distinct correlation networks between ASVs into the rhizosphere and roots of *Lotus*.** Correlation networks of ASVs in the rhizosphere (**a**) and root (**b**) compartments of plants supplemented with sources of either inorganic nitrogen (**a1, b1**) or symbiotic nitrogen (**a2, b2**) or starved (**a3, b3**) status. Only positive correlations are marked out by red lines between nodes on the networks. The nodes of the networks are colored by modularity class. Each node represents an ASV, and the size of the node is given by its degree (the degree of a node refers to the number of other nodes it is connected to). The modularity classes are denoted M1, M2, M3, M4, M5, and M6, and the numbers of ASVs in each module are shown in brackets. For each module, the proportion of ASVs at the taxonomic order level is shown by donut plots. The main taxonomic genus in each modularity is pointed out by text on the donuts (**a1, b1**) or dots below the modularity (**a2, a3, b2, b3**). The color of the dots represents the taxonomic order. The size of the dots represents the percentage of ASVs in the module.

and, importantly, that these communities differ significantly from those associated with nitrogen-fixing plants.

## Plants grown in fertilized soil have microbiomes with reduced connectivity

The large changes observed in bacterial communities due to nitrate fertilization indicate an impact on the interactions established within these communities. To investigate this possibility, we analyzed the co-occurrence patterns for bacterial ASVs from communities of plants under starved, symbiotic, and fertilized conditions. The networks identified for fertilized plants had fewer edges compared to those of symbiotic or starved plants, and the nodes were primarily isolated (Fig. 4, Supplementary Fig. 10c). Moreover, the modules contained a limited number of vertices, which in general belonged to a particular genus (Fig. 4a1, b1, Supplementary Data 4). In contrast, the observed networks of symbiotic or starved plants have more edges and over 60% of their nodes are in a module (Fig. 4a2, a3, b2, b3, Supplementary Fig. 10). Bacterial ASVs from Burkholderiales and Rhizobiales were identified as hubs in all major modules, but, interestingly, ASV1 which most likely is the symbiont was not among these (Supplementary Fig. 10, Supplementary Data 4). Importantly, the composition and taxonomic representation of ASVs within modules varied, indicating that different interactions may be established between bacteria of these communities (Fig. 4 and Supplementary Fig. 10). *Burkholderia* and *Mesorhizobium* were enriched in modules of symbiotic plants, while *Rhizobacter* and *Allorhizobium* were enriched in modules of starved plants. This indicates that symbiosis promotes an environmental niche wherein distinct members of the *Mesorhizobium* and *Burkholderia* genera may engage in positive interactions with the symbiont. This observation was confirmed by the finding that ASVs present in the same module as ASV1 (rhizosphere M3 and root M1) of symbiotic plants had a lower abundance in communities of *nfr5* and *chit5* (Fig. 3 and Supplementary Data 1, 4).

Together, the co-occurrence networks revealed that bacteria are highly interconnected and establish different interactions in communities associated with symbiotic and starved plants, while the networks

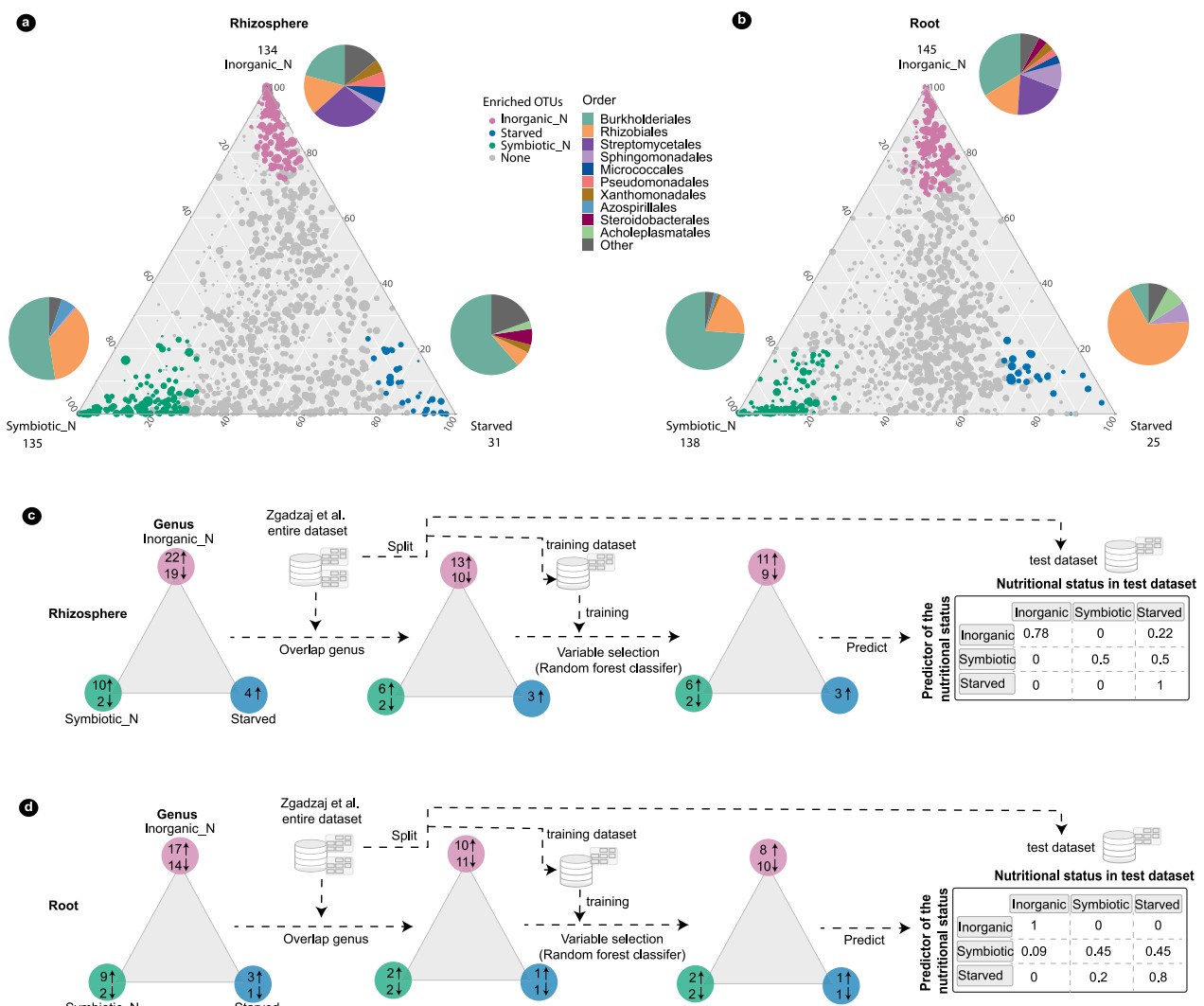

**Fig. 5 | Specific bacterial taxa enriched in the three nutritional statuses are identified as predictors with high accuracy.** Ternary plots illustrating the RA of all ASVs identified in the rhizosphere (**a**) or root (**b**) compartments of nitrogen-replete (inorganic nitrate- KNO₃ or symbiotic nitrogen) or -depleted (starved) samples. Starved conditions are represented by the *nfr5* and *chit5* grown in unfertilized Cologne soil, symbiotic nitrogen conditions are represented by the wild type and *nfre* grown in unfertilized Cologne soil, and inorganic-nitrogen conditions are represented by all genotypes grown in nitrate-supplemented Cologne soil. The ASVs are presented by dots; the size of the dots is determined by the mean RA across all three conditions in the ternary plots. The position of the dots in the ternary plot is determined by the mean RA of the ASVs within the three states. Green ASVs are enriched in symbiotic nitrogen condition, blue in starved condition, and pink in inorganic-nitrogen condition. The numbers of enriched ASVs are marked at the corners. Pie charts show the order-level taxonomic composition of the enriched ASVs. **c** Scheme of the process to identify predictor taxa in the rhizosphere and **d** root. The triangle in the scheme is a simplified symbol of the ternary plot in (**a**) and (**b**). Pink indicates inorganic-nitrogen conditions, green indicates symbiotic nitrogen conditions, and blue indicates nitrogen-starved conditions. Numbers in the circles indicate the number of enriched (upwards) or depleted (downwards) bacterial genera. The tables display the confusion matrices.

of plants grown in fertilized soil establish different and fewer interactions.

## Specific genera can predict the nitrogen nutritional status of *Lotus* with high accuracy

The observed large differences in composition and connectivity between communities associated with *Lotus* plants supported by different nitrogen-nutrition modes prompted us to determine if these are part of a more general and possibly predictable pattern. For this, we identified significantly enriched and depleted taxa associated with the three states and evaluated their potential use for predicting the nitrogen-dependent state in an unrelated/unknown microbiome of the same host grown in the same soil and under the same conditions

(Fig. 5, see also "Methods" section). First, we identified 51 and 43 genera in the rhizosphere and root, respectively, that were enriched or depleted in at least one of the three nutritional states (Fig. 5c, d, Supplementary Data 5). Next, we applied a variable selection procedure[50] to pick the most informative variables from among these genera. These were then used as covariates in several models to predict the nitrogen nutritional state of *Lotus* plants from independent but similar data set presented in Zgadzaj et al.[7]. Using a support vector machine (Fig. 5c, d), we found a prediction accuracy of 79% for the rhizosphere and 74% for the root data set. These results indicate that genera identified as being enriched or depleted in one of the three states can accurately predict the nitrogen nutritional state of *Lotus* plants.

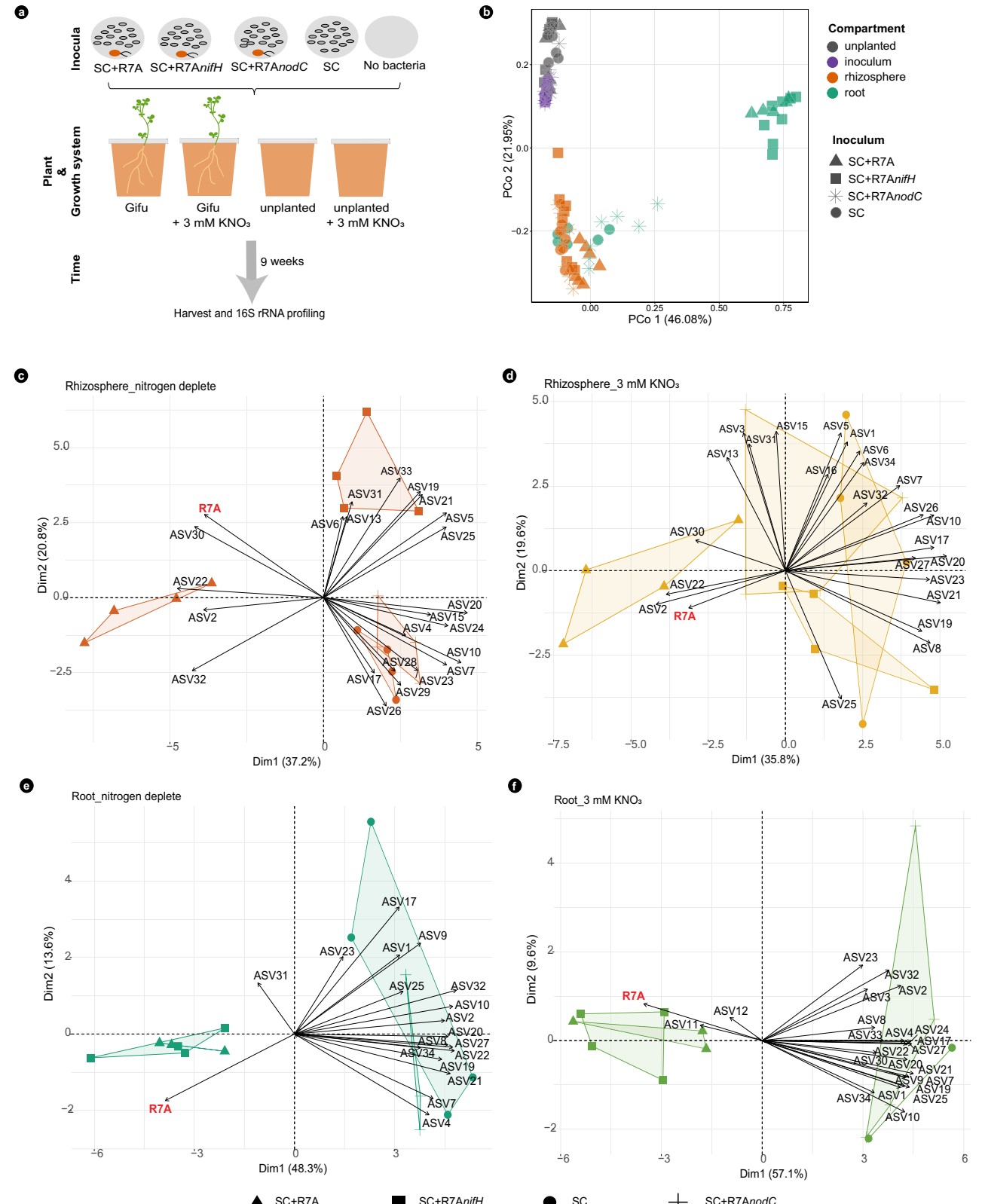

**Fig. 6 | Nod factor-producing symbiont structures *Lotus* root-associated microbiota composition. a** Experiment design. **b** PCoA analysis based on Bray-Curtis distances on all samples. Principal component analysis biplot of ASVs on rhizosphere (**c**, **d**) and root samples (**e**, **f**) from plants grown in the absence (**c**, **e**) or presence of KNO₃ (**d**, **f**). ASVs in each sample are shown as variables. The arrows point out which condition is driven by the variable. The PCA plots are shown by the first and second dimensions.

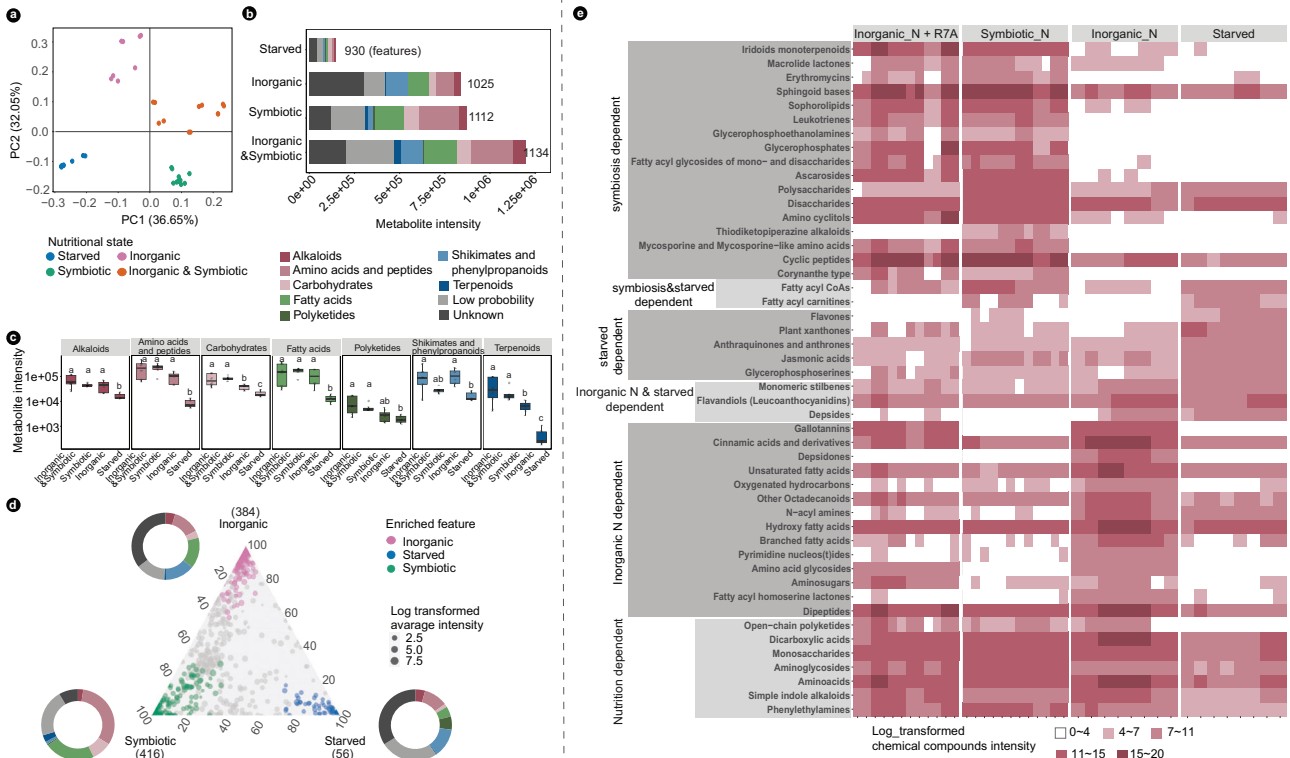

**Fig. 7 | Distinct metabolites are identified in *Lotus* root exudates grown in symbiotic, starved, or inorganic-nitrogen conditions. a** PCA analysis of the chemical features identified in root exudates; the nitrogen nutritional status of samples is marked by colors. **b** Barplot illustrates the absolute abundance of chemical features in each of the nutritional status. The mean intensity of chemical features within samples is shown. **c** Boxplot shows the abundance of metabolite intensity within the individual pathway (median, 75th percentile, and 25th percentile), and statistical analysis is conducted within the individual pathway ($p < 0.05$, $n = 6$ biologically independent samples). **d** Ternary plot illustrates statistically enriched features in the three nitrogen nutritional status, donut plots at each corner of the triangle show the composition of metabolite intensity at the pathway level. **e** Heatmap shows enriched chemical compounds identified according to the nitrogen nutritional status. The chemical compound intensity in each of the samples is illustrated by color intensity.

## Microbiota of *Lotus* is shaped by the presence of a Nod factor-producing symbiont

We found that nitrogen-fixing symbiosis has an impact on *Lotus* microbiota[7,45], but it is unknown whether nitrogen nutrition provided by the symbiont is the sole determinant for the observed differences between wild-type and symbiotic mutants (Fig. 3). Controlling the presence and functions of the symbiont in soils is not feasible; thus, we used synthetic communities and gnotobiotic experimental systems to answer this question. We assembled a taxonomically diverse "symbiont-free" synthetic community (SC) containing 61 isolates corresponding to 36 ASVs (Supplementary Data 6) from the *Lj*SPHERE[12]. This was used as a basic community to assess the impact that two critical properties of the symbiont: the capacity to produce Nod factors and the ability to fix nitrogen, have on *Lotus* microbiota. For this, we supplemented the SC with the wild-type symbiont *M. loti* R7A (SC+R7A), the Nod factor-impaired mutant *M. loti* R7A*nodC* (SC+R7A*nodC*), or the nitrogen fixation-impaired mutant R7A*nifH* (SC+R7A*nifH*). Communities associated with wild-type *Lotus* after exposure to these SCs in the presence or absence of nitrate (Fig. 6a) were analyzed and compared (Fig. 6, Supplementary Fig. 11, and Supplementary Fig. 12). Beta diversity analysis identified a significant separation of communities from the root, rhizosphere, and input samples (Fig. 6b). Root microbiomes of plants exposed to SC+R7A or SC+R7A*nifH* clustered together, irrespective of the presence of nitrate. A second cluster contained root communities of plants inoculated with R7A*nodC* and SC as well as all rhizosphere communities, while the third cluster contained the inoculum communities (Fig. 6b). This separation indicates that root communities are primarily shaped by the presence of the plant and the presence of a Nod factor-producing symbiont, while rhizosphere communities are primarily shaped by the plant. A separate analysis only of rhizosphere samples identified that communities of plants inoculated with SC+R7A separated from those exposed to other communities (Fig. 6c, d). This separation was driven by the symbiont and ASVs from *Burkholderiaceae*, *Oxalobacteraceae*, and *Beijerinckiaceae*. In the absence of nitrate, we found that rhizosphere communities of plants inoculated with R7A*nifH* were also separated from those with SC and R7A*nodC* (Fig. 6c). These results indicate that, in the absence of nitrate, R7A's production of Nod factor and its ability to fix nitrogen impact the assembly of *Lotus* rhizosphere communities. This was further confirmed in an independent experiment using a different but proficient symbiotic *Mesorhizobium* strain isolated from Cologne soil (Supplementary Fig. 13). A further investigation of communities at the taxonomic level identified that abundances of isolates from Rhizobiales, Burkholderiales, Pseudomonadales, and Xanthomonadales were significantly affected by the presence of a Nod factor-producing symbiont (Supplementary Figs. 11d and 12a). The root communities were dominated by the symbiont (average 91%) irrespective of the presence of nitrate or bacterial *NifH* gene (Fig. 6e, f and Supplementary Fig. 11e). This clear dominance of the symbiont is in marked contrast to the composition of root communities from plants grown with R7A*nodC*. Here, the relative abundance of the symbiont was reduced to 7.9%, and the root communities were enriched in members of Xanthomonadales, Burkholderiales, and Rhizobiales (Supplementary Figs. 11e and 12b).

Together, these results show that if symbiosis is enabled, the assembly of root and rhizosphere microbiota of *Lotus* is shaped by the presence of Nod factor-producing rhizobia, which provides the symbiont with a competitive advantage for root colonization and affects

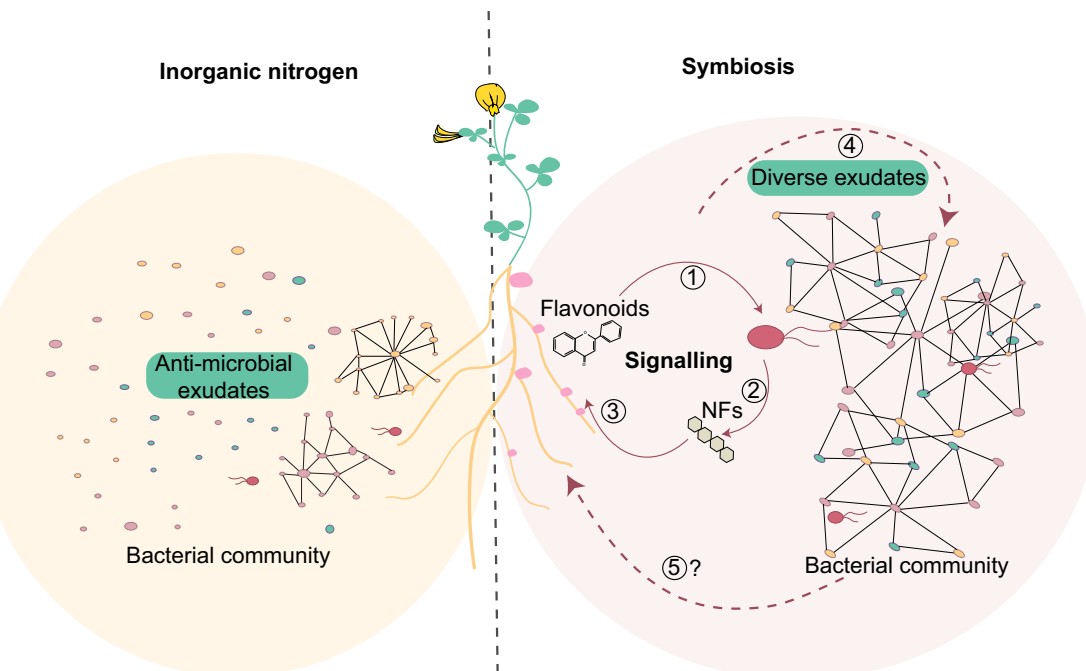

**Fig. 8 | Nitrogen nutrition and signaling during root nodule symbiosis impact the community assemblies.** *Lotus* plants grown in the presence of inorganic nitrogen secrete specific metabolites and assemble a microbial community with low connectivity. *Lotus* plants grown in symbiosis-permissive conditions secrete metabolites such as flavonoids (1) that induce Nod factor production in compatible nitrogen-fixing *Rhizobium* isolates (2). Nod factors are recognized by the *Lotus* host which initiates a signaling pathway (3) to accommodate the symbiont. Symbiotically active roots have an exudate profile (4) and associated microbial communities that differ from plants grown in the presence of inorganic nitrogen. It remains to be determined how bacterial communities associated with symbiotically active plants impact the host to promote the symbiotic association and plant growth (5).

the assembly of remaining members of the microbiota. Together, the symbiotic-permissive community provides a benefit to the plant host (Supplementary Fig. 13b).

### Exudates of *Lotus* vary in chemical composition depending on the nitrogen nutrition state

Root and rhizosphere microbiota are modulated by the exudates secreted by the plant[14–16]. Thus, we hypothesized that root microbiota identified in this study could be modulated by the exudates of *Lotus* grown under different nitrogen nutrition regimes. To test this, we grew plants either in axenic, nitrogen-starved, or nitrate-supplemented conditions or in the presence of the *M. loti* symbiont (Supplementary Fig. 14). There was an additional condition, wherein both the symbiont and nitrate were included in the assay to better differentiate between features produced under the two independent states. We detected a total of 1145 metabolite features in the exudates collected from plants grown across the four conditions (Supplementary Data 7). As predicted, the composition and intensities of metabolites varied with the nitrogen-nutrition state (Fig. 7a), mirroring the separation observed in the root and rhizosphere microbiomes (Fig. 2b, c). We identified a similar number of metabolites across conditions, except for the starved plants, where slightly fewer but at markedly lower intensities for the detected features were observed when compared to symbiotic and/or nitrate-supplemented conditions (Fig. 7b, c). This indicates that in the absence of nitrogen, *Lotus* plants maintained a diverse profile of metabolites in the exudates but down-regulated their intensities. In general, we found that carbohydrates and terpenoids had significantly higher intensities in symbiotic plants (Fig. 7c); both chemicals have signaling capacity in plant–microbe interactions[51,52]. We identified 56, 384, and 416 features specifically enriched in the exudates of plants grown in starved, symbiotic, or nitrate-supplemented conditions, respectively (Fig. 7d). Polyketides were overrepresented in starved plants (5.6% compared to 0.25% and 0.6% in nitrate and symbiotic

state, respectively), while shikimates and phenylpropanoids were enriched under both starved and nitrate conditions (13% and 14% versus 1.2% in symbiotic). Amino acids and peptides (31%), fatty acids (22%), and carbohydrates (8.8%) were more abundant under the symbiotic condition (Supplementary Data 7). A number of metabolites were specifically enriched or detected only in one of these conditions (Fig. 7e, Supplementary Fig. 15). Flavones, well-known attractants and inducers of symbiosis signaling in rhizobia[53], as well as xanthones and jasmonic acids were enriched in starved plants, indicative of a stressed, nitrogen-depleted state. The exudates of plants grown on nitrate-supplemented conditions contained metabolites with reported antimicrobial activities, such as depsidone, amino-acid glycosides, and oxygenated hydrocarbons. The metabolic profile of exudates from symbiotic plants was most diverse and varied from signaling and energy-providing chemicals such as polysaccharides, disaccharides, and fatty acyl glycosides to antimicrobial chemicals such as alkaloids and monoterpenoids.

Taken together, our analyses of metabolites revealed that *Lotus* plants adjust the panel of metabolites and their intensities in their root exudates according to a specific nitrogen regime. Importantly, we determined that plants grown in the symbiotic or nitrate-supplemented conditions differ significantly in their exudate profile, implying selection effects on the assembled bacterial communities.

## Discussion

Complex metabolic and signaling events are at the core of microbiota establishment in different ecological niches, and studies across environments and conditions suggest that a high level of taxonomic diversity is beneficial for microbiota homeostasis[54,55]. Studies in humans, mice, zebrafish, and flies have shown that perturbations in host diet have direct and predictable consequences on their gut microbiota[56–60]. Soil provides the nutrients required for plant growth, and its properties have the largest impact on rhizosphere and root

microbiota[11,61,62]. We show here that microbiota associated with *Lotus* plants provided with different sources of nitrogen (inorganic or symbiotic) are significantly different. *Lotus* plants found in a starved, symbiotic, or nitrate-replete state differ in their root and shoot transcriptomes[63] and metabolic state[64], and we show here that they exude different compounds in the rhizosphere (Fig. 7). We found that these physiological states of the plant host are associated with distinct root microbiomes and that particular genera enriched in these communities have a high level of predicting the nitrogen nutritional state of the host (Fig. 5). Nitrate fertilization induced significant changes in the metabolites produced and exuded by the plant, and changes in microbiota composition when applied to the soil but especially in the rhizosphere and root communities of *Lotus* grown in fertilized condition. A similar response of microbiota to inorganic-nitrogen application was reported for wheat and tomato[65,66]. Importantly, our analyses of the co-occurrence networks for microbiomes present in the three states of *Lotus* plants revealed the clear negative effect of inorganic nitrogen on microbial connectivity. At this stage, it is unclear whether this is a result of changes in the host metabolism and root exudate pattern (Fig. 7), a result of the functional capacities exerted by the members of these communities, or both.

Plant pathogens can also impact the root-associated microbiota by directly affecting microbiota members through the secretion of antimicrobial effectors[67]. Here, we investigated the role of the symbiont on the assembly of root microbiota and provided evidence that symbiont-derived Nod factor signals contribute to microbiota homeostasis indirectly, via the host. The symbiont modulates the remaining members of the community by inducing Nod factor-dependent signaling in the host, which in turn ensures a diverse and interconnected bacterial microbiome, likely via changes in the exudate profile (Figs. 7 and 8). Our study provides additional knowledge that increases our understanding of how the *Lotus* host exerts its crucial role on structuring the microbiota. Nod factor production is induced in symbionts primarily when host-specific (iso)flavonoids are secreted from starved roots (Fig. 7)[53]. Furthermore, the host controls the activation of Nod factor signaling in the root only when compatible Nod factors are perceived by specific LysM receptors[68]. Here we show that Nod factor signaling modulated by host genetics is part of the general control of plant-associated microbiota in a symbiosis-permissive state. Thus, the *Lotus* host is in continuous control both of its colonization by the symbiotic partner and of the associated microbiota. These findings obtained from experimental studies using plant and symbiont mutants are in line with recent results that emerged from theoretical modeling studies predicting that cooperation within the host microbiome is driven by the host's control over symbionts[69].

A symbiosis between nitrogen-fixing bacteria and legumes was found to contribute to a diverse and beneficial root-associated microbiota[7,42,43,45]. Results from analyses using bacterial mutants as well as different plant genotypes grown in nitrogen-depleted or replete conditions revealed a tight dependency of *Lotus* microbiota on the capacity of the host to mount active Nod factor signaling. Root microbiota of soil-grown plants was gradually affected, according to the degree of impairment in the Nod factor signaling present in the three mutants (Fig. 1b). A possible explanation for this pattern is that Nod factor signaling leads to changes in the physiology of the plant root, which in turn impacts associations with commensal members. Indeed, we found that the chemical profile of exudates from symbiotically active plants differs significantly from that of starved or inorganic-nitrogen-supplemented plants (Fig. 7a, d, e). The symbionts present in the soil were not identified as hubs by our network analyses but emerged as major players in structuring the communities indirectly, via a host-mediated feedback effect on the remaining members of bacterial communities. A similar feedback control of microbiota was previously described for anaerobic, beneficial commensals producing short-chain fatty acids during colonization of mice colons[56]. SCFAs

activate PPAR-γ-signaling in mice, which ensures a metabolism that preserves hypoxia in the colon environment, and thus maintains microbial homeostasis limiting the expansion of members from *Enterobacteriacea* that induce dysbiosis[70]. The *nfr5* and *chit5* plants with impaired Nod factor signaling, as well as wild-type plants inoculated with SC+R7A*nodC* or SC, are starved and have communities that differ from those of symbiotic plants (Figs. 2, 3, and 6). Interestingly, we found that in the absence of Nod factor signaling, there is an increased abundance of members from Pseudomonadales and Xanthomonadales in root microbiota of *Lotus* grown in gnotobiotic conditions (Supplementary Fig. 11e). The isolates included in our studies are commensals that don't have a detrimental impact on the host, but well-described plant pathogens emerge from these two taxa[71,72]. Thus, an increased abundance could provide increased chances for pathogenic associations to evolve or become apparent. Future studies may reveal whether Nod factor signaling limits root dysbiosis in legumes.

Collectively, our findings provide evidence that signaling established between the plant host and the compatible symbiont impacts the assembly and properties of the remaining members of the community. Furthermore, it supports the idea that Nod factor signaling established in legumes after recognition of nitrogen-fixing bacteria facilitates not only nitrogen nutrition but also the association with diverse and highly connected bacterial communities at the root–soil interface supporting nitrogen-fixing symbiosis.

## Methods
### The plant material and growth in agricultural soil for microbiota analysis
The ecotype Gifu B-129 of *L. japonicus* was used as wild-type, and the symbiosis-deficient mutants *nfre:1*, *nfre:2*[40], *chit5:1*, *chit5:2*[41], *nfr5:2*, and *nfr5:3*[29] were derived from the Gifu B-129 genotype. The *Lotus* seeds were disinfected using 1% bleach for 15–20 min, and the bleach was removed with water wash five times. Seeds were incubated on wet filter paper in Petri dishes for four days (21 °C, day/night cycle = 16/8 h). The germinated seedlings were potted in $4 \times 4 \times 9$ cm pots filled with Cologne Agriculture soil (CAS9). Two seedlings were grown in one pot to support the plants reaching the reproductive stage. Plants were grown in greenhouse conditions (day/night cycle = 14/10 h, temperature: day/night = 22/18 °C, humidity: 75% Rh) for nine weeks until all the plants started to show flower buds. Plants were watered from the bottom with either sterile water or 10 mM KNO$_3$ (Fig. 1a). The 10 mM KNO$_3$ was chosen to suppress the symbiosis of *Lotus*, and the water-supplied soil was defined as unfertilized soil for *Lotus* to establish symbiosis. Around 30 plants in 15 pots were grown for each allele in each condition (sterile water vs. 10 mM KNO$_3$). As a control, pots filled with CAS9 soil without plants were considered bulk soil.

### Sample collection and 16S rRNA amplicon sequencing of plants grown in agricultural soil
Plants grown in the same pot were considered as one sample. Three biological replicates, each with three technical replicates were analyzed for each allele in each condition. Samples of four compartments (bulk soil, rhizosphere, root, and nodules) were collected. In brief, the soil was removed from the roots until there were firmly attached soil particles on the roots, and the upper 4 cm of roots (starting 1 cm below the hypocotyl) were collected. Fractionation of rhizosphere and root was conducted by wash process (three times sterile water by 30 s vortex each time, two times detergent by 30 s vortex each time) and surface sterilization (one time 80% ethanol by 30 s vortex, one time 3% bleach by 30 s vortex). The first wash from roots was centrifuged (15 min, 4700 rpm); the pellet was collected as the rhizosphere compartment. Roots were collected as root compartment after the washing process. Nodules and visible primordia were separated from root fragments under a microscope using a scalpel. The bulk soil in the same growth conditions was collected as the soil compartment. In

parallel, the shoot fresh weight and nodule numbers of individual plants were recorded.

The collected root, rhizosphere, nodules, as well as soil samples were homogenized, and DNA was extracted using the FastDNA Spin kit for Soil (MP Bioproducts) according to the manufacturer's protocol. The variable v5–v7 region of 16S rRNA was amplified using primer pairs 799F (AACMGGATTAGATACCCKG) and 1192R (ACGTCATCCC-CACCTTCC) at the first round PCR. Indexing for distinguishing samples was done at the second round PCR using Illumina-barcoded primers targeted at the 1192R. The indexed 16S rRNA amplicons were pooled, purified, and sequenced by the Illumina Miseq platform.

### Plant growth in the gnotobiotic system for microbiota reconstitution

Magenta boxes containing lightweight expanded clay aggregate (LECA, 2–4 mm, autoclaved at 120 °C for 20 min) were used as the gnotobiotic system for two independent reconstitution experiments (Fig. 6, Supplementary Fig. 13). Individual liquid cultures of bacterial isolates (Supplementary Data 6) were cultured with tryptic soy broth (3 g l$^{-1}$, TSB, Sigma-Aldrich) liquid media for up to 72 h before inoculation (28 degree) to make sure all isolates reached the exponential growth period. Bacterial liquid cultures were washed twice by centrifuging at 4700 rpm for 15 min, discarding supernatant and resuspending with 0.25× B&D media[73]. The washed bacterial liquid cultures were pooled together accordingly, consisting of the basic community (SC) where the wild-type symbiont (*M. loti* R7A, CM: Cologne soil derived symbiotic *Mesorhizobium* strain) or mutants (*M. loti* R7A*nodC*, *M. loti* R7A*nifH*)[74] were added to establish different inocula (SC+R7A, SC+CM, SC+R7A*nodC*, SC+R7A*nifH*) for the two independent reconstitution experiments (Fig. 6, Supplementary Fig. 13). The SC and related inocula were prepared once for each reconstitution experiment. Each of the magenta boxes containing 10 disinfected wild-type seedlings was supplied with 50 ml SynComs (OD$_{600}$ = 0.02), and plants were maintained at 21 °C, day/night cycle = 16/8 h. Six magenta growth systems were established for each condition (different inocula). An aliquot of 300 µl SynComs was stored at −20 °C for sequencing as the starting inocula. All manipulation for inoculation and plotting plants was conducted on a sterile bench. The plants were kept in closed magenta boxes for the first four weeks, and then the lids of magentas were removed and the 0.25× B&D media (with or without 3 mM KNO$_3$) was supplied to the plants regularly. Note that our previous experiment using 10 mM KNO$_3$ in this gnotobiotic system had a negative impact on plant growth (yellow leaves and stunted growth were observed), thus the concentration was reduced to 3 mM KNO$_3$. After 9 weeks, the plants in some conditions reached the reproductive stage and were harvested.

### Sample collection and 16S rRNA amplicon sequencing of plants grown in the gnotobiotic system

Plants grown in the same magenta box were collected as one sample. Four replicates were collected for each condition. Samples of four compartments (unplanted, rhizosphere, root, and nodules) were collected using the same fractionation methods as plants grown in agricultural soil. The LECAs inoculated with SynComs without plants were collected as control samples.

The 16S rRNA amplicon sequencing of collected samples was done using the same procedure as described above for agricultural soil samples.

### Preprocessing of raw reads for 16S rRNA amplicons

Raw reads of 16S rRNA amplicons were processed using a method that combined QIIME[75] and USEARCH[76]. ASV clustering was conducted using the UNOISE algorithm[77]. Reads that were 97% identity to clustered ASVs were mapped as read counts to generate the ASV table. The taxonomic assignment was done with the SINTAX algorithm. The

AsCoM reference database developed on CAS soil was used as the reference for taxonomic assignment[49]. ASVs assigned as mitochondrial or chloroplast and which had a relative abundance of less than 0.01% across all the samples were removed. The filtered ASV table was used for downstream analysis.

The raw reads obtained from the reconstitution experiment were preprocessed by mapping reads back to reference sequences (USEARCH, UPARSE-REF algorithm). Reads that were at least 99% identical to reference sequences were kept for generating the ASV table and for downstream analysis.

### Plant growth in the gnotobiotic system for collecting root exudate

Twelve disinfected wild-type Gifu seedlings were grown on sterile square Petri dishes filled with 0.25× B&D media solidified with 1.4% Agar Noble (Difco). Surface of the agar media was covered with sterile filter paper. Unplanted plates were included as control. Four growth conditions of *Lotus* were established by applying *M. loti* R7A (symbiotic status), 10 mM KNO$_3$ (nitrate status), both *M. loti* R7A and 10 mM KNO$_3$ (symbiotic and nitrate status), or just media (starved status). Six replicates were conducted for each condition. Root exudate was collected after 4 weeks of growth.

### Sample collection and LC-MS/MS data analysis of root exudate

Plants and sterile filter paper were flipped into a collection system where sterile clean sand (Sigma-Aldrich) and sterile glass plates were filled in the sterile square Petri dishes (Supplementary Fig. 14), and the exudates were collected by flushing the sand with sterile water at 24 h after plants were moved into the collection system. Around 15 ml water from flushing was filter sterilized (0.45 µM) and freeze-dried to collect chemicals.

The collected samples were analyzed by an ultra-high-performance liquid chromatograph (UHPLC) coupled to a quadrupole time-of-flight mass spectrometer (qToF MS, Bruker Compact) with electrospray ionization. An untargeted analysis of metabolites was performed. The raw data were preprocessed by MzMine 3.3.0[78] (Supplementary Fig. 14) for mass detection, chromatogram building, smoothing, deconvolution, isotope grouping, feature alignment, and gap-filling to construct the feature list. Features detected at the end of the chromatogram (retention time > 19 min) were discarded. Chemical features collected in control plates were considered as a baseline and were thus subtracted from the root exudate samples. Chemical classes were assigned to metabolites with the NPC compound classification system by SIRIUS 5.6.3[79]. Detailed parameters used for the feature list construction can be found on the uploaded MzMine batch file (MassIVE, doi:10.25345/C5VQ2SM58, accession: MSV000092000).

### Statistical analyses and data visualization

The statistical analyses and most data visualizations were conducted in R v4.2.3.

### Alpha and beta diversity

The ASV table was rarefied to the lowest read numbers in the samples to calculate the Chao1 diversity indices. Significant differences were determined using the Kruskal-Wallis test (krus.test in R, $p < 0.05$). To estimate beta diversity for the samples, the ASV table was normalized using the cumulative sum scaling method[80]. Bray-Curtis distances between samples were used for constrained or non-constrained principal coordinate analysis (CPCoA, capscale function; PCoA, cmdscale function). The PERMANOVA analysis was performed with the adonis2 function from the vegan package in R.

### Differential abundance

The R-package edgeR v3.36.0 was used to fit a negative binomial regression with quasi-likelihood estimation[81] to the ASV counts. The

null-hypothesis |log2FC| ≥ 1 was tested with the glmTreat function[82]. Correction for multiple testing was carried out using the Benjamini–Yekutieli adjustment[82]. The nominal FDR level was set to 0.05. The ternary plots were constructed using ggtern v3.3.5[83].

## Phylogenetic tree, heatmap, and visualization

The ASV sequences were aligned by clustalo[84] and used to construct phylogenetic trees by raxml[85]. The phylogenetic trees were adjusted according to published phylogeny[12]. The RA of ASV was log2 transformed for heatmap visualization using the Interactive Tree of Life web tool[86].

## Co-occurrence networks

Correlations between ASV abundances were estimated using fastspar[87]. The correlation strength exclusion threshold for SparCC's iterative procedure was set to 0.2. Testing for significant correlations was performed using the permutation bootstrapping procedure implemented in fastspar (20,000 for each condition, 50,000 specifically for rhizosphere in the inorganic nitrate state). The resulting $p$-values were corrected for multiple testing using Benajmini–Hochberg with a nominal FDR level of 0.1. To equalize the means between genotypes, the function "removebatcheffect" from the package limma v3.50[88] was used. The correlations with an absolute value of less than 0.6 were discarded. The correlation networks were visualized in Gephi, and the modularity, betweenness centrality, and degree of correlation were calculated based on the correlations between ASVs.

## Prediction analysis

Prediction of the nitrogen status was carried out on the data from Zgadzaj et al.[7] using genera as predictors. Candidate covariates were identified with differential abundance analysis after collapsing the ASV data from the soil experiment into genera. The Zgadzaj et al. data set was split into a training and test data set. A third of the biological replicates from each nutritional status were assigned to test data and the rest was assigned to the training data. Independent filtering was performed by removing any genus that did not appear in at least 90% of samples. The centered log-ratio (CLR) transformation was applied to the genus abundances. A support vector machine was fitted on the training data with the covariates selected during variable selection[50]. The predictions were validated by applying the model to the test data.

## Identifying metabolites with differential intensity in the different nutritional states

The metabolite data was normalized using a procedure similar to the default option in DESeq2[89]. After the normalization, zeros were replaced by 0.5 and the data was log-transformed. Metabolites present in less than 10% of the samples were not included in the analysis. Next, we applied a linear mixed model[90,91] to identify metabolites where the effect of the nutritional state differs significantly between the Gifu samples and the control samples. Only the metabolites with $p < 0.05$ after adjusting for multiple testing with Benjamini–Yekutieli were retained for further analysis. Then, pair-wise tests between the states were conducted on the Gifu samples using another linear mixed model, and $p$-values were corrected for multiple testing with Benajmini–Hochberg.

## Reporting summary

Further information on research design is available in the Nature Portfolio Reporting Summary linked to this article.

## Data availability

All the amplicon raw data has been deposited in the National Center for Biotechnology Information NCBI database with accession number PRJNA974421. The metabolomic raw data files are stored at MassIVE Data at doi:10.25345/C5VQ2SM58 with accession number MSV000092000. Source data are provided with this paper.

## Code availability

All the scripts including detailed parameters and data to reproduce the figures for this study have been deposited in the github [https://github.com/taoke1/Project_of_nitrogen_source_and_Nod_factor_signalling].

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

## Acknowledgements

We thank Prof. Paul Schulze-Lefert and Dr. Ruben Garrido-Oter for their continuous support of our studies and critical reading of the manuscript. We thank Dr. Kathrin Wippel for constructive discussions and critical reading of the manuscript. We thank Dorthe B. Jensen for helping with the experimental set-up and Finn Pedersen for the propagation of plants. We thank Taylor Grace FitzGerald for proofreading the manuscript and the two anonymous reviewers for constructive suggestions in improving the manuscript text. This work was supported by the Bill and Melinda Gates Foundation and the UK's Foreign, Commonwealth and Development Office (FCDO) through the Engineering Nitrogen Symbiosis for Africa project (ENSA; OPP11772165), the Danish Council for Independent Research (9041-00236B), the Molecular Mechanisms and Dynamics of Plant-microbe interactions at the Root-Soil Interface project (InRoot), supported by the Novo Nordisk Foundation grant NNF19SA0059362. The China Scholarship Council supported K.T. and S.Z. for their Ph.D. study.

## Author contributions

K.T., S.Z. and A.M. performed experimental studies of the plants with help from Z.B. and S.K. I.T.J. and K.T. performed computational analysis of the data. C.L.S. and P.N.B. performed chemical analyses of root exudates. E.V.R., I.T.J. and K.T. performed statistical analyses of root exudate metabolites. S.R. and K.T. conceived the experiments. S.R., M.G., L.J. and R.W. coordinated studies. K.T., I.T.J. and S.R. wrote the manuscript with input from all authors.

## Competing interests

The authors declare no competing interests.
