## [Peer Review File · Nature Communications]

Nitrogen and Nod Factor Signaling determine *Lotus japonicus* Root Exudate Composition and Bacterial AssemblyReviewer #1 (Remarks to the Author):

There is a lot to love in this paper. It presents very interesting data, nicely-designed experiments, excellent and deep analyses, and beautiful figures. I am impressed and excited by the results that are presented here, and I think they are very important in the field because they start to connect the dots between the world of symbiosis (where we have lots of mechanistic and genetic data, from controlled but largely unrealistic conditions) versus the world of microbiomes (where we have lots of patterns from realistic but uncontrolled conditions). Particularly exciting are demonstrations that it is symbiosis per se (and even nod factor signaling) that influences plant root and rhizosphere microbiomes including rhizobial partners (both SNF partners and not!), not simply plant N status. Unfortunately the manuscript suffers from several issues that, in its current state, prevent the work from having the impact that it should. I took some time to really try and be clear about what I mean, which makes for a long review, but I hope my comments will be helpful to the authors. Thank you for the nice results, which I will surely cite in my own work. Below I outline comments both holistic and fine-scale:

Overall the introduction, discussion, and methods are each very short (~2 pages each) while the results are very long (~10 pages). I would argue that there is just too much being squished into a too-short manuscript. I do not know which aspects can be trimmed (perhaps in some sections root or rhizosphere?), but I was exhausted by the middle of the results. By the time it gets to flavonoids it's just way too many results to keep track of, at least at the level of detail that is currently in the manuscript and the lack of story-telling/integration. Perhaps some prudent moving of less-interesting things into the supplements. I think there is an art to fitting a complex and multi-year series of studies into a very short format, and I think the authors could work a bit more to strike that balance here. As written the results are very much like an ordinary paper, but the length of the manuscript overall requires a much "tighter" presentation to fit all this in.

There are several terminology issues throughout. It almost feels like the results were written by different people and then never fully pulled together/aligned. The same treatments are referred to by at least three different terms (e.g., native, natural, symbiosis-permissive – I would call these "unfertilized field soil" or unfertilized, which is very direct and clear) and it's not clear when and whether they mean the same thing. It is confusing. The methods section does not help (see comments below), and given the format comes after the results anyhow. I would strongly advocate for an approach where the intro sets up the treatments and the logic behind them, and thereafter they are simply called up by their treatments (e.g., "unfertilized") rather than receiving names that imply their meaning, e.g., "symbiosis-permissive". It gets very confusing even for a reader who is well-versed in the role of N in decreasing symbiosis.

Title: "map out" is very casual and doesn't have a well-defined scientific meaning. I would replace with a more precise verb. Determine? Influence?

Abstract:

- The rationale is not clear here. Why are these things important to know?
- "symbiosis-permissive" and "symbiosis-suppressive" do not mean anything to the reader of the abstract – use more direct terms or define
- "delineating" – again it's a little unclear what this means here

Introduction (comments by line): Overall the intro is full of facts, but is currently not a compelling or persuasive argument in support of the work presented. Details below – some comments are very minor/issues of taste – others are bigger-picture.

35: legume versus brassica? Or legume or brassica over some other plant that is not those?

44: the first paragraph is full of factual information, but does very little to set up an important knowledge gap about symbiosis or microbiomes that motivates the questions or the rest of the paper.

References 35-41 are reviews or from other species too; Lotus comes up and species-specific information at line 55. Then we're back to general information? It's confusing. I would keep it general here and introduce the model system later.

65-70: this section is imprecise and confusing. "how symbiosis signaling..." is vague as a question, and the importance of this area of study is never argued. It's so important – I think it's worth teaching the reader why this is so important.

74: again "how" here is used as a vague and imprecise substitute for a good question. To be clear, I think the paper answers very interesting and good questions, but they are simply not articulated in the introduction. Which is too bad because I think this is very cool work.

77: permissive/suppressive is not defined and thus confusing – also how does this align with the replete, starved, etc states in line 69?

82-84: Perhaps most importantly of all, the introduction would benefit immensely from strongly-framed and justified hypotheses. Directional effects are best and when it's complicated traditional hypotheses can really help your reader!: "if nod signaling is the mechanism that influences ..., then nodulation mutants will ..."

Because there are several approaches/questions, and the materials/methods come after the results, I would suggest adding a section at the end of the intro that previews the various experiments and questions. Then the reader is prepared for the whole story in the results. I think it would have helped me know where the article is going!

Results (by line):

88: this info seems to belong in the intro

90:m Here I tried to lay out the experimental design because I was confused - # genotypes x # N treatments x # reps. Some of this information is not in the methods either, though I looked.

92: native and agricultural? Again how do these labels overlap with the previous ways that the soils are described? Are these the same soil from the field, just with nitrogen added? Or are these two microbial communities totally different to start? This seems super important for the design. The rest of my review assumes that they are the same microbial communities to start and that only NO₃ was added. If not, then this would raise a whole lot of additional caveats and questions that are not addressed in the paper or in my review.

122-138: One of the most interesting results, to me, is buried here – that the genotype differences can only be detected at a much lower phylogenetic scale than what most studies address. The difference between Fig 1b and Figs 1c,d is really telling and very important for our field.

139: could this be moved into the methods and just the question be reminded here?

153: symbiotically active – another way to say plants in low N? Where is Gifu from – is the Cologne soil native for Gifu?

154: This set of results is very interesting! Very nice treatments and analyses. Figures are really nice.

161: again a preview that this approach (with the mutants) is coming, in the intro, would strengthen the arguments

171: very neat that Meso responds, since that is the N-fixer in this host. Emphasize this? Other studies of Lotus and other legumes have also found that the top endophytic bacteria in the roots or even shoots are the rhizobial symbionts. This is really important and fascinating to me.

177: again – particularly rhizobia!

186: why not unfertilized instead of native? I also think your work will be found more easily with searchable terms that everyone else uses.

205: This is G x E of the microbiome communities and this is a really nice finding as well, and one that is underplayed in my opinion.

214: "differ significantly" – but I think you can say much more than this. How are they different?

Fig. 4 is gorgeous (though a bit overwhelming): maybe for simplicity could just show the root or rhizosphere?

271: At this point, midway through the 6th page of results, I am getting a little tired. It is really a lot to keep straight. And there are several more pages to come...

300: Have you considered ending the paper before this next section? It's a possibility, I think.

304: "it is unknown" – but didn't you show this in the results presented above?

Discussion: I feel that the discussion does too much to repeat the results right now, without putting those results into the broad and compelling context that they deserve given how very careful and interesting the work is. The first part of the discussion provides background information that we should already have (if it's not in the intro already, it probably should be). I would recommend using this section to remind the reader of the main hypotheses, what you found, and then move on to use the rest of the discussion to come back to the broad context. What is true in other systems? Medicago? Others?

453: I don't think references 68-69 are properly represented here, nor does this sentence capture the current state of thinking in mutualism evolution.

Methods:

What is the provenance of Gifu? How chosen? What is LECA? What is the experimental design? How many replicates of each treatment are included? Several methods (media recipes etc) do not have references or full descriptions – it would be impossible to replicate the designs here.

Reviewer #2 (Remarks to the Author):

The manuscript describes an original work that shows evidence that Nod factor signaling leads to changes in plant growth, as already known, and impacts the root and rhizosphere microbiota and the rhizospheric exudate patterns. Similar results were found when comparing inorganic N-supplemented substrates. The authors worked with different plant genotypes and bacterial mutants in several growth assays and two Nitrogen nutrition regimes: nitrogen-depleted or repleted conditions. The N sources were the biological nitrogen fixation in legume-rhizobia symbiosis conditions or the fertilization with inorganic N. Figure 8 summarizes the central findings adequately.

The article is of significance, it combines modern techniques that explain key phenomena in the legume-rhizobium symbiosis. The amount of work done is evident, and the reported observations are mostly novel. However, this manuscript needs improvements. The amount of information is huge. There is too much data in figures and tables, mainly supplementary figures and tables. The supplementary tables do not have appropriate names; thus, when downloading them all in a bundle, it was impossible to know which table was each file. Nothing in the file name identified each table, which may seem like a detail, but it made the review quite difficult. There are no legends in the tables. Also, much methodological information is missing or scattered in the results section, making the text complex to understand without reading it many times. Please, see some comments below.

Results and Discussion sections

*Line 93, the results of the experiment in soil are mentioned, and the authors wrote "controlled

conditions”, which, in any case, should be “semi-controlled conditions” of temperature and light.

*Line 76, The species of Lotus should be mentioned (also in line 91)

*Lines 92 (also Fig. 1): The word native has to do with the place of origin of a species. In this context, native indicates that no N was added to the soil. Please consider changing or deleting the word.

*Figure 1. It is well organized. There are some details to take into account: i) Please change the “none” word set out in b, d, and e panels to “no nitrogen” or something similar; ii) the figure legend should be rewritten in such a way that the panels (a, b, c, d, e and f) are in a continued way; iii) the statement in lines 814-815 is the expected result. No novelty is shown in this figure; thus, it is suitable to use it as a Supplementary figure.

*Lines 111-114: There is no novelty in these results, and the whole subsection (line 87) could be reduced, as it is a result that complements the analysis of bacterial communities.

*Lines 120-122: no significant differences are observed in the alpha diversity of the rhizospheres of *chit5* with or without N (Fig. 2). Also, there is no reference to the asterisk in the legend of the Figure 2.

*Line 124: The clear separation of communities in the rhizosphere and root samples is not shown in Supplementary Fig. 1b. Supplementary Figures 1 c and 1d should be mentioned here.

*Supplementary Figure 1: Title refers to roots only, but the panels (a, b, c, d, e) refer to communities from soil, rhizosphere, root, and/or nodules compartments.

*Line 174: Supplementary Fig. 4 is cited before Supplementary Fig. 3 (line 208). Renumber Figures in order.

*Lines 314-317: There are no details about the Nod factor-impaired mutant *M. loti* R7AnodC (SC+R7AnodC), or the nitrogen fixation-impaired mutant R7AnifH (SC+R7AnifH) in Methodology nor any citation. Where are they from? This data should be in the Methodology section. Also, control of plants inoculated with R7A alone should be included.

*Lines 318-319: Why do the authors use 3 mM KNO₃?

*Lines 335-337: Fig. 6c, 6d, 6e, and 6f show a separation driven by R7A and other ASVs. Are the authors utterly sure that the ASV assigned to R7A is really R7A? How [With such short partial sequences]? [Also in Fig. 11 d and e]

*Supplementary Figure 13 refers to an additional experiment with SynComs also mentioned in lines 342-344. There is no description of that experiment in the whole text.

Methodology

This section was where I found the most flaws. There are several missing details and others that I could not understand. It needs to be rewritten to accompany the good results. Please see below my many comments.

1) The objective of greenhouse assays was to analyze plant growth, nodulation, and bacterial communities in different plant and soil sections. Many questions arrived to me:

Did the authors use pots? Which pot size? How many plants grew per pot? Were the seeds sterilized before planting? How many pots per treatment did the authors use? How many sets and reps? Which plant or soil sections were harvested at 9 weeks? There is a lot of information missing in this subsection. According to lines 89-95 (Results), the treatments included wild type plants (*Gifu*) and three mutants/genotypes, in Nitrogen (N) presence or absence. No mention of these treatments nor bulk soil appears in the Methodology.

2) The aim of the first gnotobiotic assay described in the article was to reconstitute the SC. Some details in this subsection:

-Line 490: Here is the first time that the medium B&D is mentioned without any indication of what medium it is (neither its name nor its composition), but later on (Line 495) Broughton and Dilworth agar medium is mentioned. The full name (and its acronym) should be mentioned the first time it is named. Also, which B&D volume was added to each magenta? When the authors wrote “1/4,” did they mean 50/4= 12.5 ml?

-Line 491-492: The authors wrote, “Note that our previous experiment using 10 mM KNO₃ in this gnotobiotic system had a negative impact on plant growth, thus the KNO₃ concentration was reduced to 3 mM.” There is no mention (or description) in the entire text of this previous gnotobiotic experiment in which 10mM KNO₃ harmed plant growth. It is not mentioned in the whole document. Please clarify that sentence, include a citation to that fact, or remove that sentence. Be careful with the latter option since it would be correct to explain why 3mM was used instead of 10mM as in the rest of the experiments. Also, I would like to know, do you have any

idea why, in "that other experiment," 10 mM KNO₃ was negative for plant growth?

-Regarding the SynComs, I did understand that the SC was prepared from a collection reported in a previous article (citation number 12, line 307-310, results section); in that citation, the authors called that collection IRL (sequence-Indexed Rhizobacterial Library). That should appear and be detailed in the Methodology section. Moreover, Table S6 indicates in its first column "LjSPHERE_no.

(IRL1)", but there is no explanation of what IRL1 means. Likewise, what is currently described in the Methodology is the "basic community" to which the bacterial symbiont or bacterial mutants were later inoculated; for this reason, I consider it helpful to define the basic community as SC (like in the results section) and mention its inoculation in the Methodology section.

-How many independent preparations of the SynComs were made? What were the selection criteria used in the design of the SC? Did the authors select for candidates with 16S-sequences that were abundant in the original communities? Do the selected candidates reflect what is expected in the natural community? In the Discussion section, it was clarified that all the isolates included in the study were commensals. It could be interesting to include some other candidates since the natural community is supposed to be diverse in habits.

-Which plant parts were harvested at 9 weeks (line 493)? Regarding the plants, were the seeds disinfected? Etc, etc etc

3) The objective of the Petri dishes assay was to obtain rhizospheric exudates for subsequent metabolomic analysis. About this subsection:

-The filter paper and Fig S14 need to be mentioned in this subsection of the Methodology. Figure S14 is very well diagrammed, and it is well understood. However, the text must describe its details, mainly about the collection system (filter paper, glass, and sand).

-Some treatments are mentioned: Petri dishes without plants and Petri dishes with plants inoculated with R7A (with and without N); the control of plants without inoculation is not mentioned in the Methodology but does appear in Fig. S14 and results (lines 367).

-Nor is it mentioned how many Petri plates were used per treatment, how many times the set was repeated, and how many plants were placed in each Petri dish. Root exudates usually have very low concentrations of metabolites, in some cases even low enough to be detected by UHPLC.

Exudates are generally obtained in systems that involve working with many plants and/or large volumes and concentrating the exudates before their analysis. Many details are missing here.

-Also, in this subsection, the authors must provide details about the inoculation of the plants. Do they inoculate the seeds, seedlings, or plants? What was the concentration per plant of the inoculum? Did the authors disinfect the seed? Did they observe nodules at the harvest stage?

4) Line 500: When the authors mention, "At the harvest stage...", What assay are they referring to? I think this entire subsection of "Sample Collection and 16S rRNA Amplicon Sequencing" should be placed following the corresponding assay or, otherwise, it should be well clarified.

In lines 501-502: "rhizosphere, root, and nodules were separated by wash process (...) and surface sterilization. Please rephrase the sentence.

In lines 504-505: regarding the sentence "The collected root, rhizosphere, nodules, as well as soil/LECA samples ...". LECA was used in the gnotobiotic SC reconstitution experiment, and there was bacterial culture and B&D medium there. Please clarify.

Only root and rhizosphere bacterial communities are mentioned in Line 118 when the collected nodules were also homogenized, and DNA was extracted using the FastDNA according to line 504.

There is no results about nodules compartment in the text. Fig S1a shows results about alpha diversity in nodules. Fig S1b shows the only clear separation within the total samples: nodules separate from the rest. Fig. S2a also shows interesting data about the bacterial community in nodules. I consider it relevant to incorporate and discuss the results of nodules samples or, otherwise, delete the nodule separation, processing, and analysis.

5) The entire subsection "Sample collection and analysis of root exudate" (from line 524) should be placed after the Petri dishes experiment. Furthermore, the procedure is unclear (lines 525-526): the authors mention, "Chemicals exuded by the roots or present in control unplanted plates onto sand grains after O/N exposure were collected by sterile water wash". Were the paper filters [with or without plants] arranged resting on the sand? How was the extraction of the exudates? Did they wash the sand with sterile water? What volume of root exudate in water did they obtain? ¿Were the collected samples analyzed by UHPLC directly without any treatment before the injection?

6) There's a severe shortage of details about the statistical analyses (Line 536). All reproducibility details are missing.

It would be better to rearrange the methodological sections for a better understanding.

The article's title does not reflect everything that has been done, for example, metabolomics.
Consider change the title

Response to Reviewers

We thank both Reviewers for their careful evaluation of our manuscript. We are very happy to see that our work is met with enthusiasm, appreciation, and great support. We have now carefully read the comments and submitted a revised manuscript text.

Reviewer 1

1. There is a lot to love in this paper. It presents very interesting data, nicely-designed experiments, excellent and deep analyses, and beautiful figures. I am impressed and excited by the results that are presented here, and I think they are very important in the field because they start to connect the dots between the world of symbiosis (where we have lots of mechanistic and genetic data, from controlled but largely unrealistic conditions) versus the world of microbiomes (where we have lots of patterns from realistic but uncontrolled conditions). Particularly exciting are demonstrations that it is symbiosis per se (and even nod factor signaling) that influences plant root and rhizosphere microbiomes including rhizobial partners (both SNF partners and not!), not simply plant N status. Unfortunately the manuscript suffers from several issues that, in its current state, prevent the work from having the impact that it should. I took some time to really try and be clear about what I mean, which makes for a long review, but I hope my comments will be helpful to the authors. Thank you for the nice results, which I will surely cite in my own work. Below I outline comments both holistic and fine-scale:

Response: We thank Reviewer1 for a very careful evaluation of our manuscript, the appreciation of our studies, results and the encouraging response. We also thank Reviewer1 for providing very helpful suggestions for improving our manuscript. We have followed your suggestions and have submitted a revised manuscript that accommodates most of these suggestions.

2. Overall the introduction, discussion, and methods are each very short (~2 pages each) while the results are very long (~10 pages). I would argue that there is just too much being squished into a too-short manuscript. I do not know which aspects can be trimmed (perhaps in some sections root or rhizosphere?), but I was exhausted by the middle of the results. By the time it gets to flavonoids it's just way too many results to keep track of, at least at the level of detail that is currently in the manuscript and the lack of story-telling/integration. Perhaps some prudent moving of less-interesting things into the supplements. I think there is an art to fitting a complex and multi-year series of studies into a very short format, and I think the authors could work a bit more to strike that balance here. As written the results are very much like an ordinary paper, but the length of the manuscript overall requires a much "tighter" presentation to fit all this in.

Response: We agree that the manuscript is data-rich and consequently, the results section can become overwhelming and difficult to follow. We have revised the manuscript text, streamlined the presentation of data in the results section and included additional explanatory text in the introduction and discussion. We have moved sections from the discussion and added additional details in the Material and methods which is now more

ample and detailed, compared to the submitted version. As a result, we believe that the body of the manuscript has a more balanced structure now.

3. There are several terminology issues throughout. It almost feels like the results were written by different people and then never fully pulled together/aligned. The same treatments are referred to by at least three different terms (e.g., native, natural, symbiosis-permissive – I would call these “unfertilized field soil” or unfertilized, which is very direct and clear) and it’s not clear when and whether they mean the same thing. It is confusing.

Response: We have revised the text and kept the terminology “unfertilized soil” in most cases delineating the experimental condition. However, since unfertilized soils can be both symbiosis-permissive and nonpermissive, depending on their nitrogen content, we kept the term “symbiosis- permissive” in those contexts when it was important to mention that it is this particular property of the soil that has an impact on communities.

4. The methods section does not help (see comments below), and given the format comes after the results anyhow. I would strongly advocate for an approach where the intro sets up the treatments and the logic behind them, and thereafter they are simply called up by their treatments (e.g., “unfertilized”) rather than receiving names that imply their meaning, e.g., “symbiosis-permissive”. It gets very confusing even for a reader who is well-versed in the role of N in decreasing symbiosis.

Response: The method section has been revised to contain more details and was adjusted as mentioned at point 3).

5. Title: “map out” is very casual and doesn’t have a well-defined scientific meaning. I would replace with a more precise verb. Determine? Influence?

Response: The title reads now “Nitrogen source and Nod factor signaling determine the assemblies of *Lotus japonicus* root bacterial communities”

6. Abstract:

- The rationale is not clear here. Why are these things important to know?
- “symbiosis-permissive” and “symbiosis-suppressive” do not mean anything to the reader of the abstract – use more direct terms or define
- “delineating” – again it’s a little unclear what this means here

Response: The rationale is presented in the second sentence that now reads: “Symbiosis impacts the assembly of root microbiota, but it is unknown how the interaction between the legume host and rhizobia impact the remaining microbiota, and whether it is independent of nitrogen nutrition.”

7. Introduction (comments by line): Overall the intro is full of facts, but is currently not a compelling or persuasive argument in support of the work presented. Details below – some comments are very minor/issues of taste – others are bigger-picture.

Response: The revised manuscript contains a revised version of the introduction where most of the points raised by the Reviewer have been included.

35: legume versus brassica? Or legume or brassica over some other plant that is not those?

Response: The study mentioned here studied legume and brassicas.

44: the first paragraph is full of factual information, but does very little to set up an important knowledge gap about symbiosis or microbiomes that motivates the questions or the rest of the paper.

Response: See the overall response to point 7.

References 35-41 are reviews or from other species too; Lotus comes up and species-specific information at line 55. Then we're back to general information? It's confusing. I would keep it general here and introduce the model system later.

Response: The above-mentioned references include mostly original literature based on studies performed in Lotus and other legumes supporting the statements.

65-70: this section is imprecise and confusing. "how symbiosis signaling..." is vague as a question, and the importance of this area of study is never argued. It's so important – I think it's worth teaching the reader why this is so important.

Response: See the overall response to point 7.

74: again "how" here is used as a vague and imprecise substitute for a good question. To be clear, I think the paper answers very interesting and good questions, but they are simply not articulated in the introduction. Which is too bad because I think this is very cool work.

Response: See the overall response to point 7.

77: permissive/suppressive is not defined and thus confusing – also how does this align with the replete, starved, etc states in line 69?

Response: This are now defined.

82-84: Perhaps most importantly of all, the introduction would benefit immensely from strongly-framed and justified hypotheses. Directional effects are best and when it's complicated traditional hypotheses can really help your reader!: "if nod signaling is the mechanism that influences ..., then nodulation mutants will ..."

Response: See the overall response to point 7.

8. Because there are several approaches/questions, and the materials/methods come after the results, I would suggest adding a section at the end of the intro that previews the various experiments and questions. Then the reader is prepared for the whole story in the results. I think it would have helped me know where the article is going!

Response: The last paragraph of the introduction includes the suggested overview. "Here, we use *Lotus japonicus* and *Mesorhizobium loti* mutants to address these questions. Experiments using unfertilized or nitrate-supplemented soil that is permissive or suppressive, respectively, for symbiosis identified that nitrogen nutrition and Nod factor signaling are major drivers of Lotus root microbiota. These findings were further confirmed by results from studies based on gnotobiotic settings with synthetic communities."

9. Results (by line):

Response: The Results section was revised and streamlined following the overall comments. We appreciate the clear questions and appreciative comments regarding our findings presented in this section. The subpoints that need to be addressed based on the current version are presented below.

90:m Here I tried to lay out the experimental design because I was confused - # genotypes x # N treatments x # reps. Some of this information is not in the methods either, though I looked.
Response: This information is now included in the Material and Methods section.

92: native and agricultural? Again how do these labels overlap with the previous ways that the soils are described? Are these the same soil from the field, just with nitrogen added? Or are these two microbial communities totally different to start? This seems super important for the design. The rest of my review assumes that they are the same microbial communities to start and that only NO₃ was added. If not, then this would raise a whole lot of additional caveats and questions that are not addressed in the paper or in my review.

Response: This information is now included in the Material and Methods section. The reviewer is correct, the start communities are the same, and only KNO₃ was added to the sterile water.

153: symbiotically active – another way to say plants in low N? Where is Gifu from – is the Cologne soil native for Gifu?

Response: We use here two conditions that inhibit symbiosis. First is genetic by the use of plant mutants that can't acquire nitrogen via symbiosis, secondly by the use of nitrate fertilization when none of the genotypes can perform symbiosis. This is presented in the results. In this and other text contexts, "symbiotically active" means plants that can perform symbiosis. Gifu originates from Japan; however, *Lotus corniculatus*, the tetraploid relative of *Lotus japonicus*, grows in areas around Cologne.

161: again a preview that this approach (with the mutants) is coming, in the intro, would strengthen the arguments

Response: We introduce the use of mutants and explain which mutants in the and the first chapter of the results section, respectively.

186: why not unfertilized instead of native? I also think your work will be found more easily with searchable terms that everyone else uses.

Response: we use unfertilized in the revised text.

205: This is G x E of the microbiome communities and this is a really nice finding as well, and one that is underplayed in my opinion.

Response: We agree that our findings have broad implications and that several findings could be extended and explained in much more detail in the text. However, this is difficult in a paper format, and we hope that readers will use both the information provided in the current text and the figures to extract valuable information.

214: "differ significantly" – but I think you can say much more than this. How are they different?

Response: The aforementioned sentence is a sum-up of the findings presented in the preceding text.

Fig. 4 is gorgeous (though a bit overwhelming): maybe for simplicity could just show the root or rhizosphere?

Response: We thank Reviewer 1 for the appreciative comments. We agree that it is also data-rich, but we find the ability to directly compare root and rhizosphere important for following the text.

271: At this point, midway through the 6th page of results, I am getting a little tired. It is really a lot to keep straight. And there are several more pages to come...

300: Have you considered ending the paper before this next section? It's a possibility, I think.

Response: We appreciate the comments from the Reviewer and took this into consideration by streamlining the text and including more information in the Material and methods section to provide a better flow of the results text.

304: "it is unknown" – but didn't you show this in the results presented above?

Response: The unknown here refers to : "... if nitrogen nutrition provided by the symbiont is the sole determinant for the observed differences between wild-type and symbiotic mutants".

10. Discussion: I feel that the discussion does too much to repeat the results right now, without putting those results into the broad and compelling context that they deserve given how very careful and interesting the work is. The first part of the discussion provides background information that we should already have (if it's not in the intro already, it probably should be). I would recommend using this section to remind the reader of the main hypotheses, what you found, and then move on to use the rest of the discussion to come back to the broad context. What is true in other systems? Medicago? Others?

Response: The discussion has been revised emphasizing as the Reviewer suggested the main findings, and in relation to the other systems when studied.

453: I don't think references 68-69 are properly represented here, nor does this sentence capture the current state of thinking in mutualism evolution.

Response: The reviewer is right the references here are wrongly placed. We have included now the appropriate reference presenting the ecological theory and the role of the host in modulating the interactions in host-associated microbiota. This was determined by theoretical modelling and validation for data on infant gut microbiota.

11. Methods:

What is the provenance of Gifu? How chosen? What is LECA? What is the experimental design? How many replicates of each treatment are included? Several methods (media recipes etc) do not have references or full descriptions – it would be impossible to replicate the designs here.

Response: a detailed version of Material and methods is included in the revised manuscript.

Reviewer2

1. The manuscript describes an original work that shows evidence that Nod factor signaling leads to changes in plant growth, as already known, and impacts the root and rhizosphere microbiota and the rhizospheric exudate patterns. Similar results were found when comparing inorganic N-supplemented substrates. The authors worked with different plant genotypes and bacterial mutants in several growth assays and two Nitrogen nutrition regimes: nitrogen-depleted or repleted conditions. The N sources were the biological nitrogen fixation in legume-rhizobia symbiosis conditions or the fertilization with inorganic N. Figure 8 summarizes the central findings adequately. The article is of significance, it combines modern techniques that explain key

phenomena in the legume-rhizobium symbiosis. The amount of work done is evident, and the reported observations are mostly novel.

Response: We thank Reviewer 2 for the appreciative words on our work.

2. However, this manuscript needs improvements. The amount of information is huge. There is too much data in figures and tables, mainly supplementary figures and tables. The supplementary tables do not have appropriate names; thus, when downloading them all in a bundle, it was impossible to know which table was each file. Nothing in the file name identified each table, which may seem like a detail, but it made the review quite difficult. There are no legends in the tables.

Response: The revised manuscript contains the Supplementary tables which have been organized, labelled, and described in the overview sections of each table.

3. Also, much methodological information is missing or scattered in the results section, making the text complex to understand without reading it many times. Please, see some comments below.

Response: The revised manuscript contains a more detailed Material and Methods chapter.

4. Results and Discussion sections

*Line 93, the results of the experiment in soil are mentioned, and the authors wrote “controlled conditions”, which, in any case, should be “semi-controlled conditions” of temperature and light.

Response: The manuscript has been revised and details of the experimental design are now included in the Material and methods.

*Line 76, The species of *Lotus* should be mentioned (also in line 91)

Response: We indicate the species name at the beginning of the text and keep *Lotus* throughout the remaining part.

*Lines 92 (also Fig. 1): The word native has to do with the place of origin of a species. In this context, native indicates that no N was added to the soil. Please consider changing or deleting the word.

Response: we changed to unfertilized soil as suggested by Reviewer 1.

*Figure 1. It is well organized. There are some details to take into account: i) Please change the “none” word set out in b, d, and e panels to “no nitrogen” or something similar; ii) the figure legend should be rewritten in such a way that the panels (a, b, c, d, e and f) are in a continued way; iii) the statement in lines 814-815 is the expected result. No novelty is shown in this figure; thus, it is suitable to use it as a Supplementary figure.

Response: We thank you for the appreciative comment on this figure. The “none” word has been changed to “unfertilized” to keep consistent with the revised text. The legend is rewritten as suggested. The manuscript has been revised and this figure is kept as a main figure to fit with the text.

*Lines 111-114: There is no novelty in these results, and the whole subsection (line 87) could be reduced, as it is a result that complements the analysis of bacterial communities.

Response: the manuscript has been revised and streamlined for a better flow.

*Lines 120-122: no significant differences are observed in the alpha diversity of the rhizospheres of *chit5* with or without N (Fig. 2). Also, there is no reference to the asterisk in the legend of the Figure 2.

Response: The corresponding sentence in the text has been corrected. The reference to the asterisk has been added to the legend.

*Line 124: The clear separation of communities in the rhizosphere and root samples is not shown in Supplementary Fig. 1b. Supplementary Figures 1 c and 1d should be mentioned here.

Response: The citation of the figure has been corrected.

*Supplementary Figure 1: Title refers to roots only, but the panels (a, b, c, d, e) refer to communities from soil, rhizosphere, root, and/or nodules compartments.

Response: We rephrased the title to avoid confusion.

*Line 174: Supplementary Fig. 4 is cited before Supplementary Fig. 3 (line 208). Renumber Figures in order.

Response: The citation order of the Figures has been carefully checked in the revised text.

*Lines 314-317: There are no details about the Nod factor-impaired mutant *M. loti* R7AnodC (SC+R7AnodC), or the nitrogen fixation-impaired mutant R7AnifH (SC+R7AnifH) in Methodology nor any citation. Where are they from? This data should be in the Methodology section. Also, control of plants inoculated with R7A alone should be included.

Response: The revised section of Materials and Methods contains now citations for the impaired mutant of R7AnodC and R7AnifH. The control of the plants inoculated with R7A alone and the plant phenotype are presented in Supplementary Figure 13.

*Lines 318-319: Why do the authors use 3 mM KNO₃?

Response: We use 3mM KNO₃ in the gnotobiotic experiments because 10 mM KNO₃ was found to be toxic for plants in this growth system. This was not the case for the soil experiments likely due to the buffering capacity of the soil matrix. We provide this information in the revised Material and Methods section.

*Lines 335-337: Fig. 6c, 6d, 6e, and 6f show a separation driven by R7A and other ASVs. Are the authors utterly sure that the ASV assigned to R7A is really R7A? How [With such short partial sequences]? [Also in Fig. 11 d and e]

Response: The sequence of V5-V7 region of 16S rRNA of R7A is unique in our designed SynComs. The strategy to assign ASV in SynCom experiments is to map back the sequencing reads to the sequences of selected isolates we already know. Thus, we are sure the ASV are assigned to R7A.

*Supplementary Figure 13 refers to an additional experiment with SynComs also mentioned in lines 342-344. There is no description of that experiment in the whole text.

Response: A detailed Materials and Methods contains this description.

5. Methodology

This section was where I found the most flaws. There are several missing details and others that I could not understand. It needs to be rewritten to accompany the good results. Please see below my many comments.

1) The objective of greenhouse assays was to analyze plant growth, nodulation, and bacterial communities in different plant and soil sections. Many questions arrived to me:

Did the authors use pots? Which pot size? How many plants grew per pot? Were the seeds sterilized before planting? How many pots per treatment did the authors use? How many sets and reps? Which plant or soil sections were harvested at 9 weeks? There is a lot of information missing in this subsection. According to lines 89-95 (Results), the treatments included wild type plants (Gifu) and three mutants/genotypes, in Nitrogen (N) presence or absence. No mention of these treatments nor bulk soil appears in the Methodology.

Response: A detailed description of the experimental setting is provided in the revised Material and methods section.

2) The aim of the first gnotobiotic assay described in the article was to reconstitute the SC. Some details in this subsection:

-Line 490: Here is the first time that the medium B&D is mentioned without any indication of what medium it is (neither its name nor its composition), but later on (Line 495) Broughton and Dilworth agar medium is mentioned. The full name (and its acronym) should be mentioned the first time it is named. Also, which B&D volume was added to each magenta? When the authors wrote “1/4,” did they mean 50/4= 12.5 ml?

-Line 491-492: The authors wrote, “Note that our previous experiment using 10 mM KNO₃ in this gnotobiotic system had a negative impact on plant growth, thus the KNO₃ concentration was reduced to 3 mM.” There is no mention (or description) in the entire text of this previous gnotobiotic experiment in which 10mM KNO₃ harmed plant growth. It is not mentioned in the whole document. Please clarify that sentence, include a citation to that fact, or remove that sentence. Be careful with the latter option since it would be correct to explain why 3mM was used instead of 10mM as in the rest of the experiments. Also, I would like to know, do you have any idea why, in “that other experiment,” 10 mM KNO₃ was negative for plant growth?

Response: Details of the media used in the experiment is presented in the revised Material and methods section. See response to point 4 regarding the use of 3mM KNO₃ in the reconstitution experiments versus 10 mM in the soil experiments.

3. Regarding the SynComs, I did understand that the SC was prepared from a collection reported in a previous article (citation number 12, line 307-310, results section); in that citation, the authors called that collection IRL (sequence-Indexed Rhizobacterial Library). That should appear and be detailed in the Methodology section. Moreover, Table S6 indicates in its first column “LjSPHERE_no. (IRL1)”, but there is no explanation of what IRL1 means. Likewise, what is currently described in the Methodology is the “basic community” to which the bacterial symbiont or bacterial mutants were later inoculated; for this reason, I consider it helpful to define the basic community as SC (like in the results section) and mention its inoculation in the Methodology section.

Response: We have revised Table S6, and removed the term “IRL1” for the confusing information here. And we have defined the basic community as SC as suggested in the Materials and Methods.

4. How many independent preparations of the SynComs were made? What were the selection criteria used in the design of the SC? Did the authors select for candidates with 16S-sequences that were abundant in the original communities? Do the selected candidates reflect what is expected in the natural community? In the Discussion section, it was clarified that all the isolates included in the study were commensals. It

could be interesting to include some other candidates since the natural community is supposed to be diverse in habits.

-Which plant parts were harvested at 9 weeks (line 493)? Regarding the plants, were the seeds disinfected? Etc, etc etc

Response: The manuscript has been revised with a detailed Materials and Methods. The information of the SynComs and the harvesting procedure are included. We include only commensals of Lotus in this SynCom. The SynCom used here is derived from the LjSphere which represent most of bacteria associated with Lotus roots when grown in natural soil. We can see the value of including other types of microbes in controlled experiments, such as pathogens, however these syncoms would be tailored to answer other questions such as resilience to pathogen infections. Moreover, we do not have in our collection a pathogenic strain associated with Lotus roots grown in Cologne soil.

3) The objective of the Petri dishes assay was to obtain rhizospheric exudates for subsequent metabolomic analysis. About this subsection:

-The filter paper and Fig S14 need to be mentioned in this subsection of the Methodology. Figure S14 is very well diagrammed, and it is well understood.

However, the text must describe its details, mainly about the collection system (filter paper, glass, and sand).

-Some treatments are mentioned: Petri dishes without plants and Petri dishes with plants inoculated with R7A (with and without N); the control of plants without inoculation is not mentioned in the Methodology but does appear in Fig. S14 and results (lines 367).

Response: The revised manuscript contains additional information in the Materials and Methods for the exudates collection system.

Nor is it mentioned how many Petri plates were used per treatment, how many times the set was repeated, and How many plants were placed in each Petri dish. Root exudates usually have very low concentrations of metabolites, in some cases even low enough to be detected by UHPLC. Exudates are generally obtained in systems that involve working with many plants and/or large volumes and concentrating the exudates before their analysis. Many details are missing here.

Response: The revised manuscript contains this information in the Materials and Methods.

-Also, in this subsection, the authors must provide details about the inoculation of the plants. Do they inoculate the seeds, seedlings, or plants? What was the concentration per plant of the inoculum? Did the authors disinfect the seed? Did they observe nodules at the harvest stage?

Response: The revised manuscript contains this information in the Materials and Methods.

4) Line 500: When the authors mention, “At the harvest stage...”, What assay are they referring to? I think this entire subsection of “Sample Collection and 16S rRNA Amplicon Sequencing” should be placed following the corresponding assay or, otherwise, it should be well clarified.

In lines 501-502: “rhizosphere, root, and nodules were separated by wash process (...) and surface sterilization. Please rephrase the sentence.

In lines 504-505: regarding the sentence “The collected root, rhizosphere, nodules, as

well as soil/LECA samples ...". LECA was used in the gnotobiotic SC reconstitution experiment, and there was bacterial culture and B&D medium there. Please clarify.

Response: The revised Materials and Methods is now revised as suggested. Detailed information has been included.

Only root and rhizosphere bacterial communities are mentioned in Line 118 when the collected nodules were also homogenized, and DNA was extracted using the FastDNA according to line 504. There is no results about nodules compartment in the text. Fig S1a shows results about alpha diversity in nodules. Fig S1b shows the only clear separation within the total samples: nodules separate from the rest. Fig. S2a also shows interesting data about the bacterial community in nodules. I consider it relevant to incorporate and discuss the results of nodules samples or, otherwise, delete the nodule separation, processing, and analysis.

Response: The nodules were found, as expected and as previously reported for *Lotus* and other legumes, dominated by the symbiont. Consequently, we did not put a large emphasis in this manuscript on describing the communities found in the nodule compartment. We mention that nodules have been analysed and included in the analysis in the manuscript text that reads now: "As expected, the diversity analysis of unplanted soil, rhizosphere, root, and nodules showed separation between the four compartments (Supplementary Figure 1b)..."

5) The entire subsection "Sample collection and analysis of root exudate" (from line 524) should be placed after the Petri dishes experiment. Furthermore, the procedure is unclear (lines 525-526): the authors mention, "Chemicals exuded by the roots or present in control unplanted plates onto sand grains after O/N exposure were collected by sterile water wash". Were the paper filters [with or without plants] arranged resting on the sand? How was the extraction of the exudates? Did they wash the sand with sterile water? What volume of root exudate in water did they obtain? Were the collected samples analyzed by UHPLC directly without any treatment before the injection?

Response: The revised Materials and Methods has changed the arrangement as suggested. Detailed information has been included.

6) There's a severe shortage of details about the statistical analyses (Line 536). All reproducibility details are missing.

Response: The materials and methods included all statistical methods used for our analyses. We agree with the reviewer that statistical analysis needs to be as detailed as possible, thus, we provided all the scripts used for the statistical analyses that can reproduce all the results presented.

6. It would be better to rearrange the methodological sections for a better understanding.

Response: A revised version of the Material and Methods section was included.

7. The article's title does not reflect everything that has been done, for example, metabolomics. Consider change the title-

Response: We agree with the Reviewer that the manuscript is data and method rich. We have chosen a title reflecting the overall findings of the manuscript.

Reviewer #1 (Remarks to the Author):

Overall I think this manuscript has been improved very much in revision. The results are much more streamlined and the discussion is more readable and features more interpretation. As before, I think the findings are quite strong, the design of the experiments is strong, the presentation of the data in the figures is great. I still think the authors are almost too humble in arguing for the importance and novelty of their work, particularly in the intro where there is still a lot of background with little persuasive argument to motivate the reader. Instead as the reader, I must bridge the gap between "this isn't known" and "this is super important because X" since the paper rarely provides this kind of context. But perhaps that is a matter of taste and overall I believe this work is very strong and should be published.

Reviewer #2 (Remarks to the Author):

I am happy to see how the manuscript structure has improved in its new version. There is a more detailed methodology section, It is easier to understand what the authors did now than before. The results have also improved since it was very difficult to complete reading the manuscript without getting lost along the way. Overall, the manuscript is more specified, clearly described, and itemized now

There is a lot of work done here. It is worth highlighting the beautiful figures, which make up the bulk of the article. The combination of metagenomics with metabolomics makes this study unique with the finding that symbiont's Nod factors activate the host's Nod factor signaling modulating root exudate profiling and root and rhizosphere bacterial assembly.

Please, consider my comments below:

- 1) Despite the authors did change the title, it does not reflect anything related to the metabolomic analysis done in this work. It does not reflect either the findings about the rhizospheric microbiomes.
- 2) Line 20: I think the word "symbiotic" should not be there. "Associated" is a more appropriate word in this context
- 3) I still think there are obvious results, like those in lines 112-114: "and that nitrate fertilization inhibited symbiosis but ensured plant nutrition independent of symbiosis impairment (Fig. 1c, f)." to even mention, or please, do not give it the importance given in the results. It is well known the effect of applied nitrate on symbiotic nitrogen fixation in legumes. It is already known that nitrate can ensure nitrogen nutrition to plants. Although it is dose-dependent, 10 mM of nitrate is enough for inhibition of nodulation in this case, in which previous gnotobiotic experiments using 10 mM KNO₃ hurt plant growth as reported in line 453.
- 4) Lines 352-353, The flavonoids are mentioned: "Nod factor production is induced in symbionts primarily when host-specific (iso)flavonoids are secreted from starved roots (Fig. 7) [68]." However, I can not see flavonoids in figure 7.
- 5) Lines 409-411, The authors mentioned a control of pots filled with CAS9 soil without plants. However, the Control of plants without inoculation must also be added.
- 6) Lines 457-458, "After 9 weeks, the plants in some conditions reached the reproductive stage and were harvested." In which treatment or condition (inocula) did not reach the reproductive stage? What happened with plants or conditions that did not reach the reproductive stage? it could be very interesting if there are differences in plant growth among inocula, and that should be remarked on.
- 7) Lines 464, the CrI of LECA in magenta with plants but without inocula is missing.

8) In Figure 1, *nfr5* does not appear in boxes c, d, e, and f. so please, delete "*nfr5*" from the following text: "...(c). The number of pink nodules (d) and a total number of nodules (e) per plant of wild type, *nfre*, *chit5*, and *nfr5* grown in unfertilized Cologne soil. The total number of nodules (f) per plant of wild type, *nfre*, *chit5*, and *nfr5* grown in Cologne soil supplemented with 10 mM KNO₃..."

9) Figure 1, What do the numbers between brackets mean? I think they refer to The number of plants analyzed for each genotype, please write it in the text.

10) Figure 8 is a fantastic graphical abstract of the article.

REVIEWERS' COMMENTS

Reviewer #1 (Remarks to the Author):

Overall I think this manuscript has been improved very much in revision. The results are much more streamlined and the discussion is more readable and features more interpretation. As before, I think the findings are quite strong, the design of the experiments is strong, the presentation of the data in the figures is great. I still think the authors are almost too humble in arguing for the importance and novelty of their work, particularly in the intro where there is still a lot of background with little persuasive argument to motivate the reader. Instead as the reader, I must bridge the gap between "this isn't known" and "this is super important because X" since the paper rarely provides this kind of context. But perhaps that is a matter of taste and overall I believe this work is very strong and should be published.

Response: We thank this Reviewer for the kind and appreciative evaluation

Reviewer #2 (Remarks to the Author):

I am happy to see how the manuscript structure has improved in its new version. There is a more detailed methodology section, It is easier to understand what the authors did now than before. The results have also improved since it was very difficult to complete reading the manuscript without getting lost along the way. Overall, the manuscript is more specified, clearly described, and itemized now

There is a lot of work done here. It is worth highlighting the beautiful figures, which make up the bulk of the article. The combination of metagenomics with metabolomics makes this study unique with the finding that symbiont's Nod factors activate the host's Nod factor signaling modulating root exudate profiling and root and rhizosphere bacterial assembly.

Response: We thank this Reviewer for the kind and appreciative evaluation

Please, consider my comments below:

1) Despite the authors did change the title, it does not reflect anything related to the metabolomic analysis done in this work. It does not reflect either the findings about the rhizospheric microbiomes.

Response: We have revised the title which now reads: "Nitrogen and Nod Factor Signaling determine Lotus japonicus Root Exudate Composition and Bacterial Assembly"

2) Line 20: I think the word "symbiotic" should not be there. "Associated" is a more appropriate word in this context

Response: The "symbiotic" refers here to the assembly of communities associated with a symbiotically proficient host.

3) I still think there are obvious results, like those in lines 112-114: "and that nitrate fertilization inhibited symbiosis but ensured plant nutrition independent of symbiosis impairment (Fig. 1c, f)." to even mention, or please, do not give it the importance given in the results. It is well known the effect of applied nitrate on symbiotic nitrogen fixation in legumes. It is already known that nitrate can ensure nitrogen nutrition to plants. Although it is dose-dependent, 10 mM of nitrate is enough for inhibition of nodulation in this case, in which previous gnotobiotic experiments using 10 mM KNO₃ hurt plant growth as reported in line 453.

Response: We agree with the reviewer, that this might be common knowledge, however, the Nature Communications journal has a broad spectrum of readers which not all might be acquainted with this and might be helpful in understanding the set-up and the findings.

4) Lines 352-353, The flavonoids are mentioned: "Nod factor production is induced in symbionts primarily when host-specific (iso)flavonoids are secreted from starved roots (Fig. 7) [68]." However, I can not see flavonoids in figure 7.

Response: We provide both figure 7 and reference #68 as support for the statement in the above sentence which addresses a more general aspect of how Nod factor production is induced in different rhizobial symbionts.

5) Lines 409-411, The authors mentioned a control of pots filled with CAS9 soil without plants. However, the Control of plants without inoculation must also be added.

Response: Previous study Zgadzaj et al 2015 showed that gnotobiotic Lotus plants do not contain a microbiota. In our conditions, the pots are kept open, and uninoculated plants will therefore reflect a microbiota that is condition-dependent rather than defined by the tested experimental variables.

6) Lines 457-458, "After 9 weeks, the plants in some conditions reached the reproductive stage and were harvested." In which treatment or condition (inocula) did not reach the reproductive stage? What happened with plants or conditions that did not reach the reproductive stage? it could be very interesting if there are differences in plant growth among inocula, and that should be remarked on.

Response: Plants that are grown in nitrogen-replete condition (nitrate or symbiotic) reach reproductive stage.

7) Lines 464, the Crl of LECA in magenta with plants but without inocula is missing.

Response: We have included uninoculated control in our experiments. When conducting library preparation for 16S rRNA amplicon sequencing on the control samples we could not obtain amplicon bands on agarose gels indicating no bacteria/microbiota contained in this condition. This is in line with previous study Zgadzaj et al 2015 which also showed that gnotobiotic Lotus plants do not contain a microbiota.

8) In Figure 1, nfr5 does not appear in boxes c, d, e, and f. so please, delete "nfr5" from the following text: "...(c). The number of pink nodules (d) and a total number of nodules (e) per plant of wild type, nfre, chit5, and nfr5 grown in unfertilized Cologne soil. The total number of nodules (f) per plant of wild type, nfre, chit5, and nfr5 grown in Cologne soil supplemented with 10 mM KNO₃..."

Response: this has been corrected in the revised manuscript.

9) Figure 1, What do the numbers between brackets mean? I think they refer to The number of plants analyzed for each genotype, please write it in the text.

Response: This is included in the revised manuscript.

10) Figure 8 is a fantastic graphical abstract of the article.

Response: We thank this reviewer for the kind comments.